# TIME-SERIES AUTOAUGMENT: DATA AUGMENTATION POLICY SEARCH FOR LONG-TERM FORECASTING

## ABSTRACT

Data augmentation is a popular regularization for addressing overfitting issues of neural networks. Recently, automatic augmentation showed strong results on image classification tasks. However, less attention had been given to automatic augmentation of time-series problems such as long-term forecasting. Toward bridging this gap, we propose an efficient, effective, and easy-to-code time-series automatic augmentation method we refer to as TSAA. We solve the associated bilevel optimization problem in two steps: a partial train of the non-augmented model for a few epochs and an iterative split process. The iterative process alternates between finding a good augmentation policy via Bayesian optimization and fine-tuning the model while pruning poor runs. Our method is evaluated extensively on challenging univariate and multivariate forecasting benchmark problems. Our results indicate that TSAA outperforms several strong baselines in most cases, suggesting it should be incorporated into prediction pipelines.

## 1 INTRODUCTION

Modern machine learning tools require large volumes of data to effectively solve challenging tasks. However, high-quality labeled data is difficult to obtain as manual labeling is costly and it may require human expertise (Shorten & Khoshgoftaar, 2019). Small datasets may lead to overfitting in overparameterized models, a phenomenon in which the model struggles with examples it has not seen before (Allen-Zhu et al., 2019). One of the effective methods to alleviate poor generalization issues is via *data augmentation* (DA). Data augmentation aims to generate artificial new examples whose statistical features match the true distribution of the data (Simard et al., 1998). In practice, DA has been shown to achieve state-of-the-art (SOTA) results in e.g., vision (Krizhevsky et al., 2012) and natural language (Wei & Zou, 2019) tasks.

Unfortunately, DA is not free from challenges. For instance, Tian et al. (2020b) showed that the effectivity of augmented samples depends on the downstream task. To this end, recent approaches explored automatic augmentation tools, where a good DA policy is searched for (Lemley et al., 2017; Cubuk et al., 2019). While automatic frameworks achieved impressive results on image classification tasks (Zheng et al., 2022) and other data modalities, problems with time-series data received significantly less attention. Toward bridging this gap, we propose in this work a new automatic data augmentation method, designed for *time-series forecasting* problems.

Time-series forecasting is a long-standing task in numerous scientific and engineering fields (Chatfield, 2000). While deep learning techniques achieved groundbreaking results on vision and NLP problems already a decade ago, time-series forecasting (TSF) was considered by many to be too challenging for deep models, up until recently (Oreshkin et al., 2020). While recent linear approaches showed interesting forecast results (Zeng et al., 2022), existing SOTA approaches for TSF are based on deep learning architectures that are structurally similar to vision models. In particular, current TSF deep models are overparameterized, and thus they may benefit from similar regularization techniques which were found effective for vision models, such as (automatic) data augmentation. Ultimately, our work is motivated by the limited availability of DA tools for time-series tasks (Wen et al., 2020).

The main contributions of our work can be summarized as follows: 1) We develop a novel automatic data augmentation approach for long-term time-series forecasting tasks. Our approach is based on a carefully designed dictionary of time-series transformations, Bayesian optimization for policy search, and pruning tools that enforce early stopping of ineffective networks. While these components

appear in existing work, their combination and adaptation to time-series forecasting was not done before, to the best of our knowledge. 2) We analyze the optimal policies our approach finds. Our analysis sheds light into the most effective transformations, and it may inspire others in designing effective data augmentation techniques for time-series data. 3) Our approach augments existing time-series forecasting baselines, and we extensively evaluated it on long-term forecasting univariate and multivariate TSF benchmarks with respect to several strong baseline architectures. We find that our framework enhances performance in most long-term forecast settings and across most datasets and baseline architectures.

## 2 RELATED WORK

**Time-series forecasting.** Recently, several neural network approaches for TSF have been proposed. Based on recurrent neural networks, DeepAR (Salinas et al., 2020) produced probabilistic forecasts with uncertainty quantification. The N-BEATS (Oreshkin et al., 2020) model employs fully connected layers with skip connections, and subsequent work (Challu et al., 2022) improved long-term forecasting via pooling and interpolation. Another line of works based on the transformer architecture (Vaswani et al., 2017) used a sparse encoder and a generative decoder in the Informer (Zhou et al., 2021), trend-seasonality decomposition in the Autoformer (Wu et al., 2021), and Fourier and Wavelet transformations in the FEDformer (Zhou et al., 2022), Recently, Pyraformer (Liu et al., 2021) significantly reduced the complexity bottleneck of the attention mechanism, PatchTST (Nie et al., 2022) exchanges the point-wise attention input with a tokenized sub-series representation. Finally, Zeng et al. (2022) propose a single-layer MLP with a larger input lookback.

**Data augmentation.** DA techniques have appeared since the early rise of modern deep learning to promote labeled image invariance to certain transformations (Krizhevsky et al., 2012). Typical image augmentations include rotation, scaling, crop, and color manipulations. Recent methods focused on modality-agnostic methods which blend linearly the inputs and labels (Zhang et al., 2018). Other works produce augmented views in the feature space (DeVries & Taylor, 2017; Verma et al., 2019). In contrast to image and text data, augmenting arbitrary time-series (TS) data have received less attention in the literature (Wen et al., 2020; Iwana & Uchida, 2021). In the review (Wen et al., 2020), the authors consider three different tasks: TS classification, TS anomaly detection, and TS forecasting. Their analysis is based on common time-series augmentation approaches such as scaling, adding noise (Um et al., 2017), window cropping or slicing, and stretching of time intervals (Le Guennec et al., 2016), dynamic time warping (Ismail Fawaz et al., 2019), perturbations of the frequency domain (Gao et al., 2020; Chen et al., 2023), and utilizing surrogate data (Lee et al., 2019). In (Smyl & Kuber, 2016), the authors discuss additional TS augmentation approaches including generating new TS using the residuals of a statistical TS (Bergmeir et al., 2016). Another technique would be to sub-sample the parameters, residuals, and forecasting from MCMC Bayesian models. The survey (Iwana & Uchida, 2021) further details a large list of TS augmentations such as jittering, rotation, time warping, time masking, interpolation and others in the context of time-series classification. Finally, the authors in (Wen et al., 2020) propose the selection and combination of augmentations using automatic approaches as a promising avenue for future research, which is the focus of the current work. We show in Fig. 1A two examples of DA policies we use.

**Automatic DA.** To avoid hand-tailored DA, recent efforts aimed for automatic tools, motivated by similar advances in neural architecture search (NAS) approaches (Zoph & Le, 2017). AutoAugment (Cubuk et al., 2019) used a recurrent controller along with reinforcement learning for the search process, yielding a highly effective but computationally intensive framework. Following works such as Fast AutoAugment employed Bayesian optimization and density matching (Lim et al., 2019). RandAugment (Cubuk et al., 2020) reduces the search space significantly by introducing stochasticity. Tian et al. (2020a) suggested partial training using augmentation-wise weight sharing (AWS). Further, recent approaches utilize gradients for the search problem, including the differentiable automatic DA (DADA) (Li et al., 2020b) and Deep AutoAugment (Zheng et al., 2022). Cheung & Yeung (2020) developed automatic DA that does not depend on the data-modality as it exploits latent transformations. In (Fons et al., 2021), the authors propose adaptive-weighting strategies which favor a subset of time-series DA for classification, based on their effect on the training loss.

## 3 BACKGROUND

Below, we briefly describe background information on Bayesian optimization and pruning approaches which we use to find the best augmentation policy and improve model training efficiency, respectively.

**Tree-structured Parzen Estimators and the Expected Improvement.** Bayesian optimization relates to a family of techniques where an objective function $f(x) : \mathbb{R}^d \to \mathbb{R}^+$ is minimized, i.e.,

$$\min_x f(x) . \tag{1}$$

In the typical setting, $f$ is costly to evaluate, its gradients are not available, and $d \leq 20$. For instance, finding the hyperparameters $(x)$ of a neural network $(f)$ is a common use case for Bayesian optimization (Bergstra et al., 2013). Unlike grid/random search, Bayesian optimization methods utilize past evaluations of $f$ to maintain a surrogate model $p(y|x)$ for the objective function $y = f(x)$. Thus, Bayesian optimization solves (1) while limiting the costly evaluations of $f$ to a minimum.

A practical realization of Bayesian optimization is given by Sequential Model-Based Optimization (SMBO) (Hutter et al., 2011). SMBO iterates between model fitting with the existing parameters (exploitation) to parameter selection using the current model (exploration). SMBO constructs a surrogate model $p(y|x)$, finds a set of parameters $x$ that performs best on the $p(y|x)$ using an acquisition function, applies $x$ on the objective function $f$ to obtain the score $y$, updates the surrogate model, and repeats the last three steps until convergence. Most SMBO techniques differ in their choice of the surrogate model and acquisition function. We will focus on Tree-structured Parzen Estimator (TPE) for the surrogate model, combined with Expected Improvement for the acquisition function. The main idea behind TPE is to model the surrogate via two distributions, $l(x)$ and $g(x)$, corresponding to model evaluations that yield positive, and negative improvement. Formally,

$$p(x|y) = \begin{cases} l(x) & y < y^* \\ g(x) & y \geq y^* \end{cases} , \tag{2}$$

where $y^*$ is a threshold score, and the surrogate model is obtained via Bayes rule. It can be shown that maximizing $l(x)/g(x)$ leads to an optimal Expected Improvement (EI) (Bergstra et al., 2011).

**Asynchronous Successive Halving.** While Bayesian optimization uses a minimal number of evaluations of $f$, the overall minimization is computationally demanding due to the high cost of $f$, e.g., if $f$ is a neural network that needs to be trained. To alleviate some of these costs, Asynchronous Successive Halving (ASHA) (Jamieson & Talwalkar, 2016; Li et al., 2020a) enforces early stopping of poorly performing parameters $x$, whereas parameters with low $l(x)$, are trained to the fullest. In a fixed budget system, given a maximum resource $R$, minimum resource $r$, and a reduction factor $\eta$, ASHA works as follows. One creates model checkpoints during the training process at epochs $\eta^j$ where $j = 1, \ldots, \lfloor \log_\eta R/r \rfloor$. Each checkpoint is referred to as a *rung*, and at the end of each rung, one keeps only the best $\frac{1}{\eta}$ runs. To avoid waiting for all runs to reach the next rung, ASHA performs asynchronous evaluations to promote or halt runs on the go. We illustrate in Fig. 1B an example of a baseline model with multiple different runs, administered by the ASHA policy.

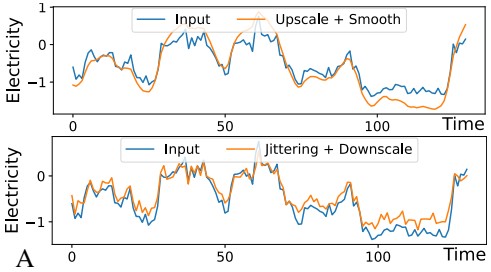 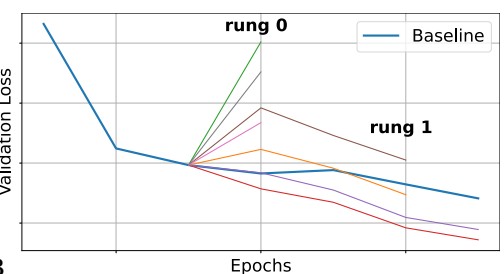

Figure 1: A) Two examples of sub-policies applied on Electricity data. B) The above plot demonstrates the behavior of ASHA with respect to the baseline model (blue). Some of the poorly performing runs are discontinued at the end of rungs, whereas the other runs train to completion.

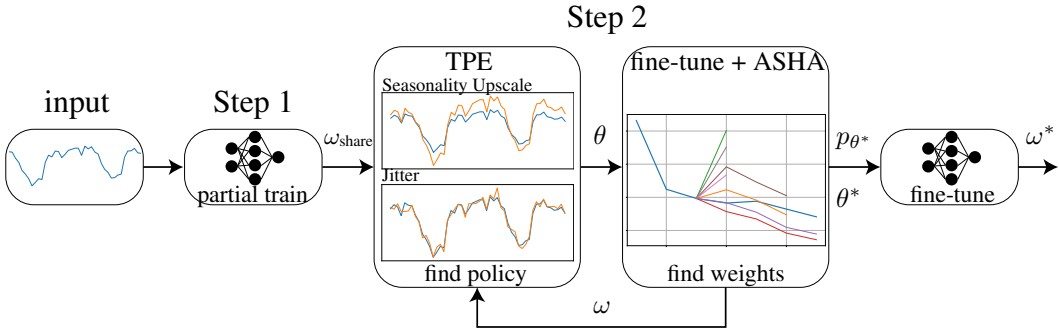

Figure 2: Our time-series automatic augmentation (TSAA) approach is based on a partial train of the model (Step 1), followed by an iterative process (Step 2) where we alternate between improving the augmentation policy $\theta$ to training the model weights $\omega$. We find $\omega^*$ by fine-tuning over $p_{\theta^*}$.

## 4  TIME-SERIES AUTOAUGMENT (TSAA)

**Automatic augmentation via bi-level optimization.**  The task of finding data augmentations automatically during the training of a deep neural network model can be formulated as a bi-level optimization problem, see e.g., (Li et al., 2020b). Namely,

$$\min_{\theta} \quad \mathcal{L}_{\text{val}}(\omega, \theta) \tag{3}$$

$$\text{subject to} \quad \min_{\omega} \mathbb{E}_{p_\theta} \left[ \mathcal{L}_{\text{tr}}(\omega, \theta) \right] \ , \tag{4}$$

where $\mathcal{L}_{\text{tr}}$ and $\mathcal{L}_{\text{val}}$ denote the train and validation losses, respectively, typically mean squared error for TSF. The parameters $\omega$ and $\theta \sim p_\theta$ correspond to the network weights and the augmentation policy. The above minimization is difficult to solve, and thus we relax it as detailed next.

**TSAA overview.**  Our approach, which we call time-series automatic augmentation (TSAA), consists of two main steps, as illustrated in Fig. 2 and summarized in Alg. 1 in App. D. In the first step, we partially train the model for a few epochs and construct a set of shared weights. The second step iterates between solving Eq. (3) in search of an augmentation policy using TPE and EI to solving Eq. (4) with fine-tuning and ASHA for an optimal model. A complexity analysis is given in App. F.

**Step 1: compute shared weights.**  Solving Eq. (4) iteratively requires repeated trainings of the deep model, which is computationally prohibitive. To reduce these costs, we propose to partially train the baseline model and generate a shared set of weights $\omega_{\text{share}}$. Doing so, Step 2 is reduced to an iterative process of fine-tuning models for a small number of epochs, where $\omega_{\text{shared}}$ are shared across all augmentations policies. Beyond efficiency aspects, applying DA in the later stages of training is assumed to be more influential (Tian et al., 2020a). In practice, we partially train for $\lfloor \beta K \rfloor$ epochs, where $\beta = 0.5$ is a hyperparameter and $K$ is the *active* number of train epochs. In our tests, $K \leq 10$, and it may be strictly less due to an early stopping scheduler. $K$ is found by training the baseline model with no augmentation to completion and saving the weights after every epoch. Then, we define

$$\omega_{\text{shared}} := \omega(\lfloor \beta K \rfloor) \ , \quad R := K - \lfloor \beta K \rfloor \ ,$$

where $R$ is the maximum resource parameter, and $r = 1$ is the minimum resource, see Sec. 3.

**Step 2: iterative split optimization.**  Given $\omega_{\text{shared}}$, it remains to solve Eqs. (3) and (4) to find the best augmentation policy $\theta^*$ and final weights $\omega^*$. In TSAA, we propose to split this problem to an iterative process, where we alternate between exploring augmentation policies $\theta$ via Eq. (3) to exploiting the current policy and produce model weights $\omega$ via Eq. (4). Namely, for a fixed set of weights $\omega$, the upper minimization finds the next policy $\theta$ to try by evaluating the validation set. Then, we fine-tune the model using a fixed $\theta$ with early stopping for a maximum of $R$ epochs to produce the next $\omega$. This procedure is repeated until a predefined number of trials $T_{\max}$ is reached. The $k$ best-performing policies define $p_{\theta^*}$ from which $\theta^*$ is sampled, where we only allow policies that improve the baseline validation loss. Finally, we fine-tune the model again to obtain $\omega^*$.

**Solving Eq.** (3). Existing work solved the upper problem using reinforcement learning (Cubuk et al., 2019; Tian et al., 2020a), grid search (Cubuk et al., 2020; Fons et al., 2021), and one-pass optimization (Li et al., 2020b; Zheng et al., 2022). Inspired by (Lim et al., 2019), we propose to use Tree-structured Parzen Estimator (TPE) with Expected Improvement (EI), see Sec. 3. In the context of TSAA, the parameters $x$ in Eq. (1) represent the policy $\theta$ and $f$ is $\mathcal{L}_{\text{val}}$. The Bayesian optimization is conducted over the policy search space and time-series augmentations we describe below.

**Policy search space.** The augmentation policies $\theta$ we consider are drawn from a distribution $p_\theta$ over $k$ sub-policies $\Theta = \{\theta_1, \ldots, \theta_k\}$, i.e.,

$$\theta \sim p_\theta := \{p(\theta_j)|\theta_j \in \Theta\} . \tag{5}$$

Each sub-policy $\theta_j$ is composed of $n$ transformations $T_{j,i}$, applied sequentially on the output data $x_{i-1}$ of the previous transformation with $x_0$ being the input data and $m_{j,i}$ being the magnitude of the transformation. That is,

$$\theta_j = T_{j,n}(x_{n-1}, m_{j,n}) \circ \cdots \circ T_{j,1}(x_0, m_{j,1}) . \tag{6}$$

**Time-series data augmentations.** While natural images are invariant to geometric transformations as translation and rotation, arbitrary time-series data need not be invariant to a certain type of transformations. Moreover, capturing the invariance in regression problems such as TSF may be more challenging than in classification tasks including images. Finally, time-series data may include slow and fast phenomena such as bursts of electricity usage and seasonal peaks, for which some DA may be inapplicable. Thus, we propose to exploit DA that manipulate some features of the data and leave some features unchanged. For example, adjusting the trend while keeping the seasonality and noise components unaffected, or diversifying the time intervals in a way that the series mean and variance still stay the same. In particular, we suggest the following time-series transformations: identity, jittering, trend scaling, seasonality scaling, scaling, smoothing, noise scaling, flip, permutation, reverse, dynamic-time-stretching (DTS), window warping, and mixup. The magnitude of the augmentations can be controlled using a single parameter. The transformations are further elaborated in App. C and Tab. 3 in the appendix.

**Solving Eq.** (4). Finally, solving the bottom minimization may be achieved in a straightforward fashion via fine-tuning. However, as motivated in Sec. 4, doing so iteratively is costly. To prune runs, we augment our approach with Asynchronous Successive Halving (ASHA). Our choice to use ASHA over other techniques such as Bayesian Optimization HyperBand (BOHB) (Falkner et al., 2018) is motivated by the following reasons. First, BOHB has shown to be slightly inferior to ASHA (Li et al., 2020a). Second, In our setting $R \in \{1, 2, .., 5\}$ and $\eta$ is set to be more aggresive. As a result, only two SHA brackets at most can be exploited in the HyperBand, thus limiting its effectiveness.

## 5 RESULTS

In what follows, we provide details regarding our experimental setup and we evaluate our approach. In the supplementary material, we give additional information on models and datasets (App. B), hyperparameters (App. E), and extended results (App. G.3).

### 5.1 IMPLEMENTATION DETAILS

**Baselines.** We train all models based on the implementation and architecture details as they appear in (Oreshkin et al., 2020) for N-BEATS and (Zhou et al., 2021; Wu et al., 2021; Zhou et al., 2022) for the Transformer-based models. The model weights are optimized with respect to the mean squared error (MSE) using the ADAM optimizer (Kingma & Ba, 2015) with an initial learning rate of $10^{-3}$ for N-BEATS and $10^{-4}$ for Transformer-based models. The maximum number of epochs is set to 10 allowing early-stopping with a patience parameter of 3. The reported baseline results are obtained using our environment and hardware, and they may slightly differ from the reported values for the respective methods. Every experiment is run on three different seed numbers, and the results are averaged over the runs. The Pytorch library (Paszke et al., 2019) is used for all model implementations, and executed with NVIDIA GeForce RTX 3090 24GB.

**Method.** We use Optuna (Akiba et al., 2019) for the implementations of TPE and ASHA. The number of trials $T_{\max}$ is set to 100. For TPE, In order to guarantee aggressive exploration at the beginning, we run the first 30% of trials with random search. For ASHA, $r$ and $\eta$ are set globally to 1 and 3 respectively. The maximum resource parameter $R$, representing the epochs, is set differently for each experiment, due to the baseline's early-stopping.

After the augmentation policy search is finalized, a maximum of $k$ best policies are selected to obtain $p_{\theta^*}$, where $k = 3$, and the final model is *fine-tuned* with $\theta^* \sim p_{\theta^*}$ using the shared weights $\omega_{\text{share}}$. We opt to fine-tune the model and not re-train from random weights so that the final model training matches our optimization process as close as possible. Indeed, Cubuk et al. (Cubuk et al., 2020) discuss the potential differences between the final model behavior in comparison to the performance of the intermediate proxy tasks, i.e., the models obtained during optimization. As the similarity in performance between these models and the final model is not guaranteed, a natural choice is to similarly train the proxy tasks and the final model, as we propose to do.

**Augmentations.** Each transformation includes a different increasing or decreasing magnitude range which are all mapped to the range $[0, 1]$. This way, $m = 0$ implies the identity and $m = 1$ is the maximum scale. To eliminate cases of the identity being repeatedly chosen, we replace the lower bound in the range with an $\epsilon > 0$ such that for all transformations in the search space only $m > 0$ is possible. The transformations *Trend scale* and *Seasonality scale* require computing the seasonality and trend components; we pre-compute these factors using the decomposition in STL (Cleveland et al., 1990) and treat it as part of the input data. Each augmentation is applied before the input is fed to the model, namely, on the input $x$ and the target $y$ of the train data batches.

## 5.2 MAIN RESULTS

In our experiments, we employ a similar setup to (Wu et al., 2021; Zhou et al., 2022), where the input length is 96 and the evaluated forecast horizon corresponds to 96, 192, 336, or 720. For ILI, we use input length 36 and horizons 24, 36, 48, 60. For a fair comparison, we re-produce all baseline results

Table 1: Multivariate long-term time-series forecasting results on six datasets in comparison to five baseline models. Low MSE and MAE values are better, and high relative improvement MSE% and MAE% scores are better. Boldface text highlights the best performing models.

| | | Informer | | Autoformer | | FEDformer-w | | FEDformer-f | | TSAA | | | |
| | | MSE | MAE | MSE | MAE | MSE | MAE | MSE | MAE | MSE↓ | MAE↓ | MSE%↑ | MAE%↑ |
|---|---|---|---|---|---|---|---|---|---|---|---|---|---|
| ETTm2 | 96 | 0.545 | 0.588 | 0.231 | 0.310 | 0.205 | 0.290 | 0.189 | 0.282 | **0.187** | **0.274** | **1.058** | **2.837** |
| | 192 | 1.054 | 0.808 | 0.289 | 0.346 | 0.270 | 0.329 | 0.258 | 0.326 | **0.255** | **0.314** | **1.163** | **3.681** |
| | 336 | 1.523 | 0.948 | 0.341 | 0.375 | 0.328 | 0.364 | 0.323 | 0.363 | **0.304** | **0.350** | **5.882** | **3.581** |
| | 720 | 3.878 | 1.474 | 0.444 | 0.434 | 0.433 | 0.425 | 0.425 | 0.421 | **0.398** | **0.403** | **6.353** | **4.276** |
| Electricity | 96 | 0.336 | 0.416 | 0.200 | 0.316 | 0.196 | 0.310 | 0.185 | 0.300 | **0.183** | **0.297** | **1.081** | **1.000** |
| | 192 | 0.360 | 0.441 | 0.217 | 0.326 | 0.199 | 0.310 | 0.201 | 0.316 | **0.195** | **0.309** | **2.010** | **0.323** |
| | 336 | 0.356 | 0.439 | 0.258 | 0.356 | 0.217 | 0.334 | 0.214 | 0.329 | **0.208** | **0.323** | **2.804** | **1.824** |
| | 720 | 0.386 | 0.452 | 0.261 | 0.363 | 0.248 | 0.357 | 0.246 | 0.353 | **0.238** | **0.348** | **3.252** | **1.416** |
| Exchange | 96 | 1.029 | 0.809 | 0.150 | 0.281 | 0.151 | 0.282 | **0.142** | **0.271** | 0.143 | 0.272 | -0.704 | -0.369 |
| | 192 | 1.155 | 0.867 | 0.318 | 0.409 | 0.284 | 0.391 | 0.278 | 0.383 | **0.270** | **0.378** | **2.878** | **1.305** |
| | 336 | 1.589 | 1.011 | 0.713 | 0.616 | **0.442** | **0.493** | 0.450 | 0.497 | 0.459 | 0.504 | -3.846 | -2.231 |
| | 720 | 3.011 | 1.431 | 1.246 | 0.872 | 1.227 | 0.868 | **1.181** | **0.841** | 1.213 | 0.842 | -2.710 | -0.119 |
| Traffic | 96 | 0.744 | 0.420 | 0.615 | 0.384 | 0.584 | 0.368 | 0.577 | 0.361 | **0.565** | **0.352** | **2.080** | **2.493** |
| | 192 | 0.753 | 0.426 | 0.670 | 0.421 | 0.596 | 0.375 | 0.610 | 0.379 | **0.571** | **0.351** | **4.195** | **6.400** |
| | 336 | 0.876 | 0.495 | 0.635 | 0.392 | 0.590 | 0.365 | 0.623 | 0.385 | **0.584** | **0.359** | **1.017** | **1.644** |
| | 720 | 1.011 | 0.578 | 0.658 | 0.402 | 0.613 | 0.375 | 0.632 | 0.388 | **0.607** | **0.368** | **0.979** | **1.867** |
| Weather | 96 | 0.315 | 0.382 | 0.259 | 0.332 | 0.269 | 0.347 | 0.236 | 0.316 | **0.180** | **0.256** | **23.729** | **18.987** |
| | 192 | 0.428 | 0.449 | 0.298 | 0.356 | 0.357 | 0.412 | 0.273 | 0.333 | **0.252** | **0.311** | **7.692** | **6.607** |
| | 336 | 0.620 | 0.554 | 0.357 | 0.394 | 0.422 | 0.456 | 0.332 | 0.371 | **0.296** | **0.355** | **10.843** | **4.313** |
| | 720 | 0.975 | 0.722 | 0.422 | 0.431 | 0.629 | 0.570 | 0.408 | 0.418 | **0.382** | **0.395** | **6.373** | **5.502** |
| ILI | 24 | 5.349 | 1.582 | 3.549 | 1.305 | **2.752** | 1.125 | 3.268 | 1.257 | 2.760 | **1.123** | -0.291 | **0.178** |
| | 36 | 5.203 | 1.572 | 2.834 | 1.094 | **2.318** | **0.980** | 2.648 | 1.068 | 2.362 | 0.984 | -1.898 | -0.408 |
| | 48 | 5.286 | 1.594 | 2.889 | 1.122 | 2.328 | 1.006 | 2.615 | 1.072 | **2.264** | **0.988** | **2.749** | **1.789** |
| | 60 | 5.419 | 1.620 | 2.818 | 1.118 | 2.574 | 1.081 | 2.866 | 1.158 | **2.520** | **1.062** | **2.098** | **1.758** |

Table 2: Univariate long-term time-series forecasting results on five datasets in comparison to five baseline models. Low MSE and MAE values are better, and high relative improvement MSE% and MAE% scores are better. Boldface text highlights the best performing models.

| | | Informer | | Autoformer | | FEDformer-f | | N-BEATS-I | | N-BEATS-G | | TSAA | | | |
|---|---|---|---|---|---|---|---|---|---|---|---|---|---|---|---|
| | | MSE | MAE | MSE | MAE | MSE | MAE | MSE | MAE | MSE | MAE | MSE↓ | MAE↓ | MSE%↑ | MAE%↑ |
| ETTm2 | 96 | 0.085 | 0.225 | 0.123 | 0.270 | **0.068** | 0.198 | 0.080 | 0.213 | 0.080 | 0.210 | **0.068** | **0.192** | 0.000 | 3.030 |
| | 192 | 0.130 | 0.282 | 0.141 | 0.289 | **0.096** | 0.238 | 0.103 | 0.240 | 0.110 | 0.250 | **0.096** | **0.237** | 0.000 | 0.420 |
| | 336 | 0.161 | 0.314 | 0.170 | 0.319 | **0.138** | **0.286** | 0.162 | 0.312 | 0.172 | 0.320 | 0.139 | 0.290 | -0.725 | -1.399 |
| | 720 | 0.221 | 0.373 | 0.206 | 0.353 | 0.189 | **0.335** | 0.199 | 0.347 | 0.201 | 0.353 | **0.187** | 0.336 | 1.058 | -0.299 |
| Electricity | 96 | 0.261 | 0.367 | 0.454 | 0.508 | **0.244** | 0.364 | 0.326 | 0.402 | 0.324 | 0.397 | **0.244** | **0.354** | 0.000 | 2.747 |
| | 192 | 0.285 | 0.386 | 0.511 | 0.532 | **0.276** | 0.382 | 0.350 | 0.417 | 0.363 | 0.420 | 0.277 | **0.368** | -0.362 | 3.665 |
| | 336 | 0.324 | 0.417 | 0.739 | 0.651 | 0.347 | 0.432 | 0.393 | 0.440 | 0.392 | 0.443 | **0.310** | **0.394** | 4.321 | 5.516 |
| | 720 | 0.632 | 0.612 | 0.673 | 0.610 | 0.408 | 0.473 | 0.458 | 0.490 | 0.489 | 0.502 | **0.378** | **0.447** | 7.353 | 5.497 |
| Exchange | 96 | 0.490 | 0.554 | 0.149 | 0.308 | 0.133 | 0.284 | 0.210 | 0.344 | 0.223 | 0.351 | **0.093** | **0.236** | 30.075 | 16.901 |
| | 192 | 0.790 | 0.721 | 0.290 | 0.415 | 0.292 | 0.419 | 1.130 | 0.840 | 0.783 | 0.675 | **0.215** | **0.352** | 25.862 | 15.181 |
| | 336 | 2.146 | 1.223 | 0.708 | 0.662 | **0.477** | **0.532** | 1.587 | 1.047 | 2.622 | 1.266 | 0.532 | 0.572 | -11.530 | -7.519 |
| | 720 | 1.447 | 1.008 | 1.324 | 0.892 | 1.304 | 0.882 | 0.870 | 0.747 | 2.588 | 1.303 | **0.527** | **0.594** | 39.425 | 20.482 |
| Traffic | 96 | 0.262 | 0.348 | 0.266 | 0.372 | 0.210 | 0.318 | 0.181 | 0.268 | 0.159 | 0.240 | **0.158** | **0.239** | 0.629 | 0.417 |
| | 192 | 0.294 | 0.376 | 0.272 | 0.379 | 0.206 | 0.311 | 0.177 | 0.263 | 0.181 | 0.264 | **0.160** | **0.243** | 9.605 | 7.605 |
| | 336 | 0.308 | 0.390 | 0.261 | 0.374 | 0.217 | 0.322 | 0.180 | 0.271 | **0.155** | **0.239** | 0.156 | 0.244 | -0.645 | -2.092 |
| | 720 | 0.364 | 0.440 | 0.269 | 0.372 | 0.243 | 0.342 | 0.226 | 0.316 | 0.212 | 0.304 | **0.189** | **0.279** | 10.849 | 8.224 |
| Weather | 96 | 0.005 | 0.048 | 0.009 | 0.078 | 0.009 | 0.073 | 0.003 | 0.044 | 0.003 | 0.043 | **0.001** | **0.024** | 66.667 | 44.186 |
| | 192 | 0.004 | 0.051 | 0.009 | 0.068 | 0.007 | 0.067 | 0.004 | 0.046 | 0.004 | 0.047 | **0.001** | **0.027** | 75.000 | 41.304 |
| | 336 | 0.003 | 0.043 | 0.006 | 0.058 | 0.006 | 0.062 | 0.004 | 0.048 | 0.005 | 0.054 | **0.002** | **0.035** | 33.333 | 18.605 |
| | 720 | 0.004 | 0.049 | 0.007 | 0.063 | 0.006 | 0.060 | 0.004 | 0.049 | 0.004 | 0.048 | **0.002** | **0.034** | 50.000 | 29.167 |

on our system, and the augmentations are applied on the same generated batches as the baseline. Our main results are summarized in Tab. 1 and Tab. 2 including all the baseline results and TSAA. For TSAA, we include the best performing model trained on all baseline architectures. The full results for every architecture with and without TSAA are provided in the appendix spanning tables 6-14. We detail the mean absolute error (MAE) and mean squared error (MSE) (Oreshkin et al., 2020). Lower values are better, and boldface text highlights the best performing model for each dataset and metric. For TSAA, we also include the *relative improvement* percentage, i.e., $100 \cdot (e_b - e_n)/e_b$, where $e_b$ is the best baseline error and $e_n$ is our result. We denote by MSE% and MAE% the relative improvement of MSE and MAE, respectively. A higher improvement is better.

**Multivariate time-series forecasting results.** Based on the results in Tab. 1, we observe that most datasets benefit from automatic augmentation, where in the vast majority of cases, TSAA improves the baseline scores. It is apparent that TSAA yields stronger performance in particular in the long-horizon settings with **6.35%** ($0.425 \rightarrow 0.398$) reduction in ETTm2, **3.25%** ($0.246 \rightarrow 0.238$) reduction in Electricity, and **2.1%** ($2.328 \rightarrow 2.264$) reduction in ILI. One of the more prominent results appears for Weather 96 and 336 with reductions in MSE of **23.73%** ($0.236 \rightarrow 0.180$) and **10.84%** ($0.332 \rightarrow 0.296$), respectively. For the Exchange dataset, TSAA obtains slightly higher errors with respect to the FEDformer-w baseline. Overall, TSAA achieves the *best results* in 39 error metrics, in comparison to FEDformer-f and FEDformer-w with 4 and 5 best models, respectively.

**Univariate time-series forecasting results.** Similar to the multivariate results, most long horizon settings benefit from TSAA. With a $21.74\%$ average reduction across all datasets with a horizon of 720. Furthermore, the results that stand out the most are the MSE and MAE reductions in Weather, with a $66\%, 75\%, 33, 50\%\%$, and respectively $44.2\%, 41.3\%, 17.6\%, 29.2\%$ performance improvements corresponding to the $96, 192, 336$ ,and $720$ horizons. Further, it is evident in Tabs. 10-14, that the improvements in the Weather dataset are not limited to a specific baseline architecture. In contrast to the multivariate setting, TSAA achieves significantly better scores on the Exchange dataset with average improvements of $21\%$ and $11.27\%$ for the MSE and MAE metrics. Notably, the results in the univariate case are slightly more involved than the multivariate setting such that that only Weather always benefits from TSAA, whereas the results for other datasets are mixed. Still, TSAA shows a positive advantage over all baseline models. In particular, TSAA obtained the best models for 32 error metrics, whereas FEDformer-f and N-BEATS-G improved 9 and 2 measures.

**Policy analysis.** The most noticeable selected transformations are illustrated in Fig. 3. It is evident that the transformations Trend Downscale, Jittering, Mixup, and Smoothing are some of the prominent

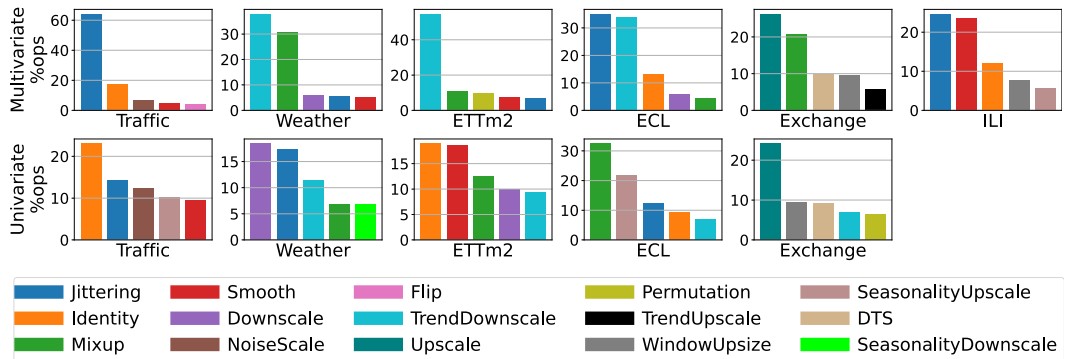

Figure 3: The best five performing transformations per dataset attained with TSAA, measured with the percentage proportion of the selected operations (%ops). Each colored bar represents a transformation and the y-axis represents the percentage proportion the given transformation accounts for.

selections in the overall setup. Trend Downscale accounts for more than $30\%$ of the operations in ETTm2, Weather and Electricity; this may indicate that the deep models tend to overestimate the trend, and thus it requires downscaling. Jittering and Smoothing on the other hand, do not violate time-series characteristics such as trend or seasonality but still promote diversity within the given dataset, where Smoothing is approximately the opposite of Jittering. Notably, Mixup appeared as one of the five most important transformations for four and three datasets in the multivariate and univariate settings, respectively. We believe that Mixup is beneficial to TSF since it samples from a vicinal distribution whose variability is higher than the original train set. We show in Fig. 4 the outcome with and without TSAA compared to the ground truth, showing that employing custom policies per signal may significantly improve forecasting.

## 6 ABLATION AND ANALYSIS

### 6.1 PARAMETER SELECTION

**Choice of $\beta$.** In what follows, we motivate our choice for the $\beta$ hyperparameter which dictates for how many epochs we pre-train the baseline architecture to obtain $\omega_{shared}$. To this end, we investigate the effect of utilizing different values of $\beta$. We consider four different settings: 1) full training with augmentation, i.e., $\beta = 0.0$, 2) half training with augmentation, i.e., $\beta = 0.5$, 3) augmentation applied only in the last epoch, 4) baseline training with no augmentation, i.e., $\beta = 1.0$. We used TSAA on the ILI with respect to N-BEATS-G in the univariate setting, and Informer, Autoformer and FEDformer-f in the multivariate case, as well as on multivariate ETTm2 with Autoformer and FEDformer-f. We plot the averaged results of these architectures in Fig. App. 6A, showing four colored curves corresponding to the various forecasting horizons $24/96$, $36/192$, $48/336$ and $60/720$ with colors blue, orange, green and red, respectively. The best models are obtained for $\beta = 0.0$ and $\beta = 0.5$, that is, full- and half-augmented training. Somewhat surprisingly, two of the four best forecasting horizons $(36/192, 48/336)$ are obtained for $\beta = 0.5$. Overall, the fully augmented model (i.e., $\beta = 1.0$) attains a **5.1%** average improvement over the baseline, whereas using $\beta = 0.5$ yields a **5.3%** average improvement. Thus, fully training with augmentation provides no improvement to the overall performance, while requiring significantly more resources. Indeed, Tian et al. (2020a) employs a similar strategy, and thus, we propose to generate $\omega_{shared}$ after training for *half* of the active epochs, and we fine-tune the model using optimal augmentation policies.

**Reduction factor and linked operations.** In Sec. 3, we introduced the reduction factor $\eta$ that controls the number of kept runs in ASHA. Additionally, we discuss in Sec. 4 that every sub-policy is composed of $n$ linked operations of time-series augmentations. Here, we would like to empirically justify our choices for these two hyperparameters. Our ablation study uses the ILI dataset on the $36, 48, 60$ forecasting tasks, with N-BEATS-G for the univariate case, and Informer, Autoformer, and FEDformer-f for the multivariate configuration. We test the values $\eta \in \{2, 3\}$ and $n \in \{1, 2\}$. Every experiment is repeated three times, and we analyze the average results.

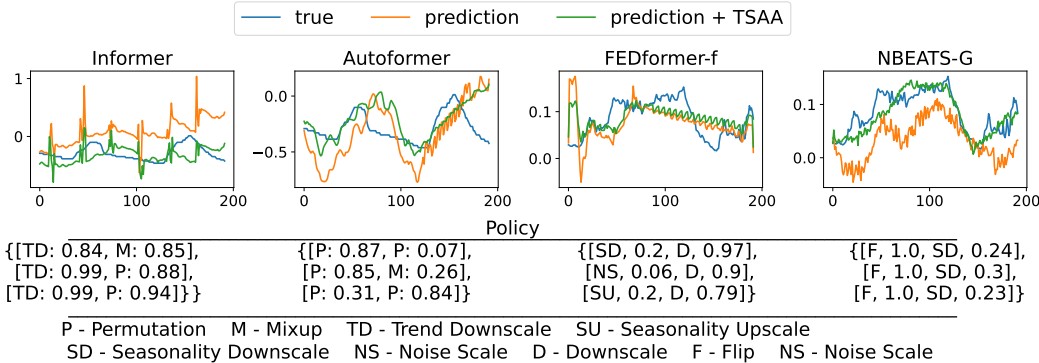

Figure 4: The ground truth, prediction, and prediction with augmentation attained with TSAA applied to the same forecast target in ETTm2 with Informer, Autoformer multivariate, and Weather with FEDformer-f and NBEATS-G univariate. It is shown that augmentation can assist the different models to achieve more accurate predictions. The attained policies are given underneath each plot.

Overall, we propose to use the values $\eta = 3$ and $n = 2$ due to the following observations arising from our experiments. The improvement difference between $\eta = 2$ and $\eta = 3$ is only **0.12%** in favor of $\eta = 2$, thus suggesting that neither exhibits a statistically-dominant performance advantage. Nevertheless, $\eta = 3$ is resource efficient as it reduces the amount of kept runs $1/\eta$ by $16.67\%$. Moreover, a single operation $n = 1$ attains a **6.4%** average improvement compared to the baseline, whereas two linked operations $n = 2$ yield a **7.4%** average improvement.

**Convergence of TSAA.** In our experiments, we look for good augmentation policies for $T_{\max} = 100$ iterations. Here, we explore the effect of this value on the performance of the resulting models. We evaluate our framework on the ILI dataset with the architectures Informer, Autoformer, FEDformer-f, N-BEATS-G and N-BBEATS-I using varying values for $T_{\max} \in \{100, 150, 200, 250\}$. Intuitively, greater $T_{\max}$ values may result in an improved convergence and a better overall performance as the framework can explore and exploit a larger variety of configurations from the search space. Indeed, we show in Fig. App. 6B the normalized average MSE values obtained for the various tests. We observe an MSE reduction of **1%** for the transformer-based models when increasing $T_{\max} = 100$ to $T_{\max} = 250$. The N-BEATS architecture benefited more and achieved a **7.25%** reduction. In conclusion, the hyperparameter $T_{\max}$ presents a natural trade-off to the practitioner: higher $T_{\max}$ values generally lead to better performance at a higher computational cost, whereas lower values are less demanding computationally but present inferior performance.

# 7 DISCUSSION

In this work, we study the task of data augmentation in the setting of time-series forecasting. While recent approaches based on automatic augmentation achieved state-of-the-art results in image classification tasks, problems involving arbitrary time-series information received less attention. Thus, we propose a novel time-series automatic augmentation (TSAA) method that relaxes a difficult bilevel optimization. In practice, our framework performs a partial training of the baseline architecture, followed by an iterative split process. Our iterations alternate between finding the best DA policy for a given set of model weights, to fine-tuning the model based on a specific policy. In comparison to several strong methods on multiple univariate and multivariate benchmarks, our framework improves the baseline results in the majority of prediction settings.

In the future, we would like to explore better ways for relaxing the bilevel optimization, allowing to train an end-to-end model (Li et al., 2020b; Zheng et al., 2022). Further, we believe that our approach would benefit from stronger time-series augmentation transformations. Thus, one possible direction forward is to incorporate learnable DA modules, similar in spirit to filters of convolutional models.

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

## A  APPENDIX

In what follows, we present additional details regarding models and datasets used in our evaluation (App. B), the transformations used in our method (App. C), algorithm pseudocode (App. D), hyperparameter values (App. E), complexity analysis of TSAA (App. F), a comparison of TSAA and other prominent AutoAugment methods (App. G.1), and lastly, we provide extended result tables for every baseline alongside TSAA (App. G.3).

## B  MODELS AND DATASETS

We extensively evaluate the performance of our Time-Series AutoAugment (TSAA) framework. To this end, we selected some of the most recent prominent time-series forecasting models. We consider the baseline architectures: **N-BEATS** (Oreshkin et al., 2020), a deep neural architecture based on backward and forward residual links and a very deep stack of fully-connected layers. **Informer** (Zhou et al., 2021) adapts the Transformer (Vaswani et al., 2017) architecture to time-series forecasting tasks, with a new attention mechanism. **Autoformer** (Wu et al., 2021) exchanges the self-attention module for an auto-correlation mechanism and introduces time-series decomposition as part of the model's encoding. Finally, **FEDformer** (Zhou et al., 2022) enables capturing more important details in time-series through frequency domain mapping.

For each of the given baseline models, we apply TSAA on six commonly-used datasets in the literature of long-term time-series forecasting: (1) **ETTm2** (Zhou et al., 2021) contains electricity transformer oil temperature data alongside 6 power load features. (2) **Electricity** (Zhou et al., 2021) is a collection of hourly electricity consumption data over the span of 2 years. (3) **Exchange** (Lai et al., 2018) consists of 17 years of daily foreign exchange rate records representing different currency pairs. (4) **Traffic** (Zhou et al., 2021) is an hourly reported sensor data containing information about road occupancy rates. (5) **Weather** (Zhou et al., 2021) contains 21 different meteorological measurements, recorded every 10 minutes for an entire year. (6) **ILI** (Wu et al., 2021) includes weekly recordings of influenza-like illness patients.

We summarize in the table below the different datasets and their attributes such as their sampling *frequency*, *variates* which determine the number of channels in each example, the total number of *timesteps* in each dataset, the different *horizon* lengths used for forecasting, and lastly the *lookback* period which is the input length used for the prediction.

| Dataset Summary | | | | | |
|---|---|---|---|---|---|
| dataset | frequency | variates | total timesteps | horizon | lookback period |
| ETTm2 | 15 minutes | 7 | 69,680 | 96, 192, 336, 720 | 96 |
| Electricity | hourly | 321 | 26,304 | 96, 192, 336, 720 | 96 |
| Exchange | daily | 8 | 7,588 | 96, 192, 336, 720 | 96 |
| Traffic | hourly | 862 | 17,544 | 96, 192, 336, 720 | 96 |
| Weather | 10 minutes | 21 | 52,696 | 96, 192, 336, 720 | 96 |
| ILI | weekly | 7 | 966 | 24, 36, 48, 60 | 36 |

## C    TIME-SERIES TRANSFORMATIONS

In this section, we offer a detailed description of the different time-series transformations, which can be found in Tab. 3 and depicted in Fig. 5 with $m = 0.85$ compared to the original signal. For each of the transformations *Trend scale*, *Scale*, *Seasonality scale*, *Window warping*, we use two separate and independent transformations to demonstrate an increase or decrease of the given effect.

Table 3: Our search space is composed of the following time-series transformations and their associated magnitude range.

| Transformation | Description | Range of magnitudes |
| --- | --- | --- |
| Jittering | Adds white noise with $\sigma$ controlled by $m$ (Um et al., 2017). * | [0,0.1] |
| Trend scale | Multiplies the trend component by $m$. * | [1,10], [0,1] |
| Scale | Multiplies the entire series by $m$ (Um et al., 2017). | [1,3], [0.3,1] |
| Seasonality scale | Multiplies the seasonality component by $m$. | [1,3], [0,1] |
| Smooth | Performs low-pass filtering with a convolution kernel, where $m$ controls the kernel size. | [0,11] |
| Noise scale | Performs high-pass filtering with a second-order convolution kernel to extract the difference, which is then multiplied by $m$ and added back to the original series. | [0,1] |
| Permutation | Exchanges two non-overlapping time intervals, such that the interval size is controlled by $m$. (Um et al., 2017). | [0,0.3] |
| Dynamic time stretching | Manipulates the length of different non-overlapping time intervals, where $m$ controls the manipulation magnitude. (Nguyen et al., 2020). | [1,5] |
| Window warping | Manipulates the length of the entire window (Um et al., 2017). | [1,1.5], [0.5,1] |
| Mixup | Linearly interpolates between two series, $m$ controls the contribution of each series (Zhang et al., 2018). | [0,0.5] |
| Identity | Returns the original series. | None |
| Flip | Flip the series relative to the value location by multiplying by $(-1)$ (Iwana & Uchida, 2021). * | {0,1} |
| Reverse | Change the relative location of the time steps to span from end to start. | {0,1} |

* marks a transformation implemented with min-max scaling to ensure equal relative changes.

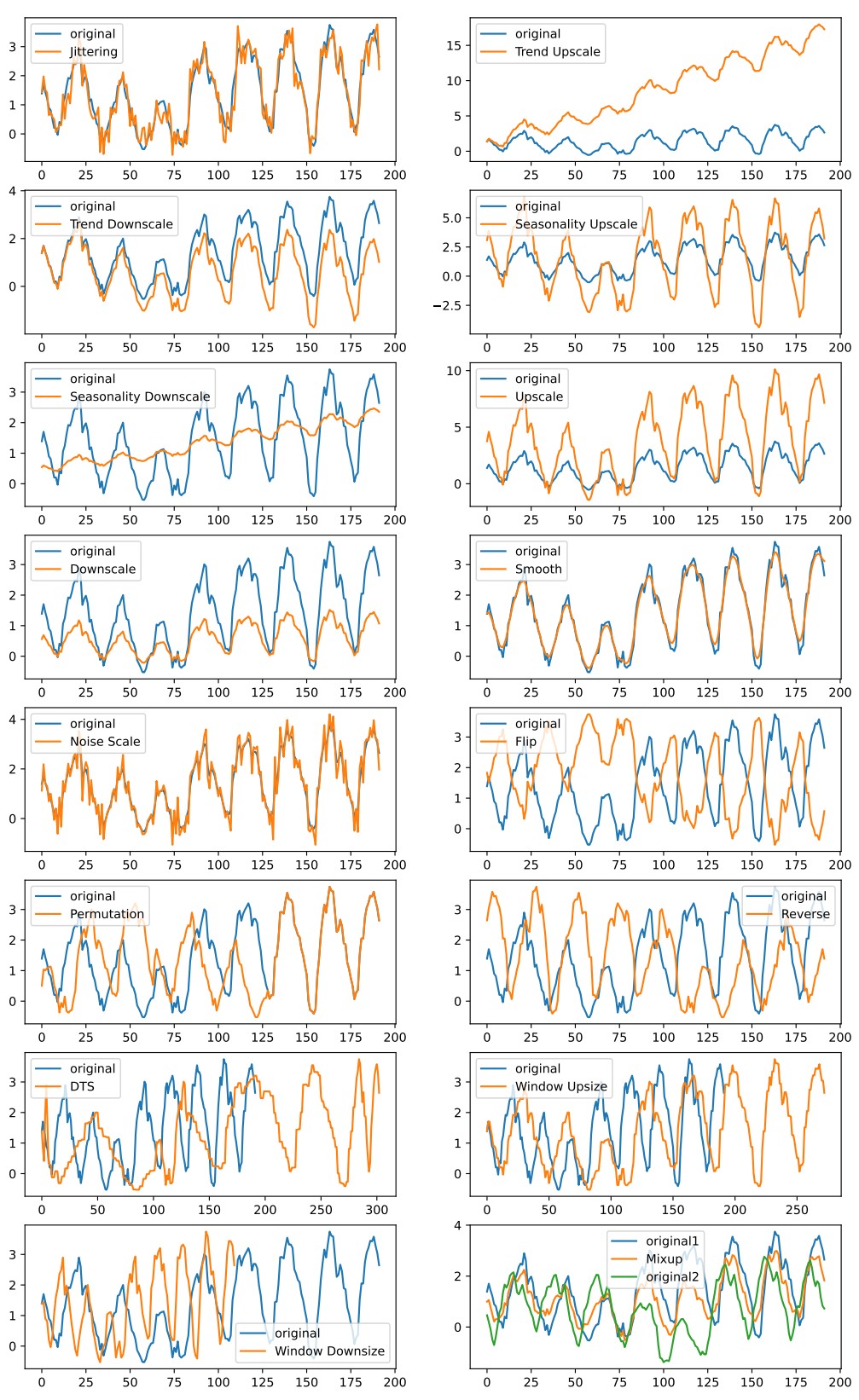

Figure 5: We demonstrate the effect of different transformations when applied to the same example from the Electricity dataset. Blue and orange represent the original signal and its transformed version, respectively.

## D  TSAA PSEUDOCODE

---
**Algorithm 1** Time-Series AutoAugment (TSAA)

---
**Inputs:** partial train factor $\beta$, resources $R, r$, max trials $T_{\text{max}}$, reduction factor $\eta$, and $k$ best DA sub-policies

$\Lambda \leftarrow \emptyset$
$\omega_{\text{share}}, R \leftarrow$ partial solve Eq. (4) with $\beta$
**for** $i = 1$ **to** $T_{\text{max}}$ **do**
    $\theta_i \leftarrow$ solve Eq. (3) with TPE$(\Theta, \omega_{i-1})$
    $w_i \leftarrow$ fine-tune Eq. (4) with $\omega_{\text{share}}$ and ASHA$(r, R, \eta)$
    $\Lambda \leftarrow \Lambda \cup \left\{ [\theta_i, \ \mathcal{L}_{\text{val}}(\omega_i, \theta_i)] \right\}$
**end for**
$p_{\theta*} \leftarrow k$ best sub-policies $\theta_i$ from $\Lambda$
$\omega^* \leftarrow$ fine-tune $\mathcal{L}_{\text{tr}}(\theta^* \sim p_{\theta*})$
**return** $p_{\theta*}, \theta^*, \omega^*$

---

## E  TSAA HYPERPARAMETERS

We detail in Tab. 4 the hyperparameter values we used in the evaluation of TSAA.

Table 4: Hyperparameter values used in the evaluation of TSAA.

| | Model Parameters - TSAA | | | | | | | |
|---|---|---|---|---|---|---|---|---|
| $T_{max}$ | exploration trials fraction | $m$ range | $n$ | $\eta$ | $r$ | resource type | $\beta$ | $k$ |
| 100 | 0.3 | (0,1] | 2 | 3 | 1 | epoch | 0.5 | 3 |

## F  COMPLEXITY

A straightforward upper bound of our method is given by $\mathcal{O}((1 - \beta)KT_{max})$ evaluated in epochs, where $K$, $(1 - \beta)$, and $T_{max}$ correspond to the number of active epochs the model trains for, the fraction of $K$ to be used for fine-tuning, and the maximum number of trials, respectively. However, for TSAA to practically reach such an upper bound when $K > 2$ would require each trial to outperform the preceding trials, thus avoiding being pruned by ASHA. This setup is very unlikely for the following reasons: (1) a fraction size $0.3$ of $T_{max}$ of starting trials are manually dedicated to random search to promote aggressive exploration at the start. (2) To the best of our knowledge Bayesian Optimization does not guarantee monotonic improvement and inherently promotes exploration (Bergstra et al., 2011). (3) $\mathcal{L}_{\text{val}}(\theta, w^*)$ is not promised to be convex with respect to $\theta$, making it even more difficult to attain a monotonic improvement. In our empirical evaluations, we have observed different operations with different transformations being selected, as opposed to a single policy being repeatedly selected, which strengthens our claim. Further, we would like to share a solid example from our empirical experiments with FEDformrer-w applied to Electricity with a horizon of 336. This setup is considered resource expensive as one epoch may take as long as 20 minutes on a single RTX3090. In the given setup, $K$ was set to 8 and $\beta = 0.5$ as defined in the hyperparameter table Tab. 4. Therefore, according to the bound mentioned above, the total number of epochs would be 400 epochs. However, the use of ASHA with Bayesian optimization reduced it to 170, proving the computational impact. For comparison, applying a naive approach without ASHA and $\beta$ will result in a total number of 800 training epochs.

## G  EXTENDED RESULTS

In this section, we provide the performance of TSAA for each individual model featured in Tab. 1 and Tab. 2 in the main text. We can observe from the multivariate results that Informer benefits the most from TSAA, with the 720 horizon in particular. On the other end, TSAA struggles to achieve significant improvements in FEDformer-f, especially for the Exchange dataset. In the univariate setting, unlike the multivariate setting, the models NBEATS-G and NBEATS-I gain large

improvement rates in Exchange thanks to TSAA. Overall, we can conclude that all models and datasets exhibit a degree of improvement with the support of TSAA.

## G.1 AUTOAUGMENT METHOD COMPARISON

We compare TSAA to other efficient AutoAugment methods via an experiment that shows that both Fast AutoAugment (Lim et al., 2019) and RandAugment (Cubuk et al., 2020) are *inconsistent* and thus inferior to TSAA. In this experiment, we tested the performance of deploying Fast AutoAugment and RandAugment with the same search space, with the exception of discretized magnitude ranges so RandAugment falls in line with the original method. The given methods were tested on Autoformer and FEDformer-f multivariate together with datasets: ETTm2, Traffic, Weather, and ILI. The results are provided in 5. For Fast AutoAugment we set $K = 3$ folds which control the number of subsets, each subset explores and exploits 100 augmentation trials. For RandAugment, we discretize the magnitude range to 8 bins and utilize the partial train scheme with the same $\beta$ as in TSAA, to allow RandAugment to benefit from the same approach. While it is shown that Fast AutoAugment and RandAugment are superior on Weather and ILI respectively, they attain inferior results on the other datasets. Nevertheless, TSAA is shown to be most effective on Traffic and ETTm2 and second-best on ILI and Weather. Additionally, TSAA maintains a consistent improvement across all datasets providing a 3.33% average MSE reduction as opposed to RandAugment which offers approximately half of that, or Fast AutoAugment with a negative average MSE reduction. In conclusion, TSAA is more consistent with better overall performance when compared to prominent AutoAugment methods in the time-series domain.

Table 5: Comparison of automatic augmentation approaches including TSAA, Fast AutoAugment and RandAugment. We denote by %AI the average improvement in percentage.

| | Metric | Best Baseline MSE | MAE | Fast AA MSE | MAE | RandAugment MSE | MAE | TSAA MSE | MAE |
|---|---|---|---|---|---|---|---|---|---|
| ETTm2 | 96 | 0.189 | 0.282 | 0.197 | 0.279 | 0.192 | 0.282 | **0.187** | **0.274** |
| | 192 | 0.258 | 0.326 | 0.262 | 0.319 | 0.257 | 0.323 | **0.255** | **0.314** |
| | 336 | 0.323 | 0.363 | 0.322 | 0.356 | 0.326 | 0.364 | **0.311** | **0.350** |
| | 720 | 0.425 | 0.421 | 0.415 | 0.407 | 0.427 | 0.420 | **0.406** | **0.403** |
| Traffic | 96 | **0.577** | **0.361** | 0.655 | 0.410 | 0.590 | 0.376 | **0.577** | 0.362 |
| | 192 | 0.610 | 0.379 | 0.652 | 0.408 | 0.622 | 0.390 | **0.601** | **0.371** |
| | 336 | 0.623 | 0.385 | 0.674 | 0.421 | 0.626 | 0.392 | **0.619** | **0.383** |
| | 720 | **0.632** | **0.388** | 0.705 | 0.427 | 0.643 | 0.396 | **0.632** | **0.388** |
| Weather | 96 | 0.236 | 0.316 | **0.191** | **0.252** | 0.203 | 0.275 | 0.207 | 0.285 |
| | 192 | 0.273 | 0.333 | **0.240** | **0.290** | 0.267 | 0.332 | 0.252 | 0.311 |
| | 336 | 0.332 | 0.371 | **0.290** | **0.321** | 0.328 | 0.364 | 0.313 | 0.355 |
| | 720 | 0.408 | 0.418 | **0.363** | **0.367** | 0.398 | 0.413 | 0.382 | 0.395 |
| ILI | 24 | 3.268 | 1.257 | 4.671 | 1.603 | 3.160 | 1.234 | **3.150** | **1.219** |
| | 36 | 2.648 | 1.068 | 3.835 | 1.375 | **2.457** | **1.019** | 2.578 | 1.049 |
| | 48 | 2.615 | 1.072 | 3.694 | 1.359 | **2.558** | **1.045** | 2.609 | 1.069 |
| | 60 | 2.866 | 1.158 | 3.855 | 1.410 | **2.775** | **1.108** | 2.805 | 1.140 |
| %AI | | 0.0 | 0.0 | -9.5 | -4.88 | 1.67 | 1.22 | **3.33** | **3.1** |

## G.2 BETA TRIALS

We show in Fig. 6 the ablation study plots related to the choice of $\beta$ hyperparameter (Fig. 6A) and convergence (Fig. 6B) we discussed in the main text in Sec. 6.

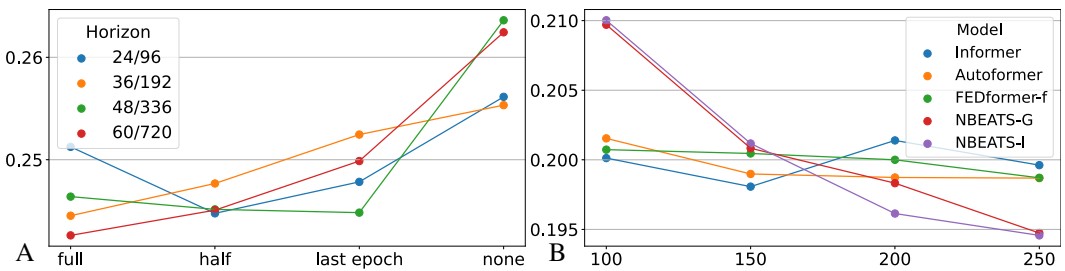

Figure 6: A) The normalized average performance measures as a function of different $\beta$ values. Our results indicate that $\beta = 0.5$ (half) attains the best computational resources to performance gain ratio. B) We plot the normalized average performance measures as a function of different $T_{\max}$ values. As the number of trials grows, we observe better overall performance.

## G.3 TSF RESULTS

Tabs. 6, 7, 8, 9, 10, 11, 12, 13, 14 detail the extended results for every baseline architecture and dataset we considered in the main text.

Table 6: Informer multivariate

| | | Informer | | TSAA | | | |
| | | MSE | MAE | MSE↓ | MAE↓ | MSE%↑ | MAE%↑ |
|---|---|---|---|---|---|---|---|
| ETTm2 | 96 | 0.545 | 0.588 | **0.224** | **0.321** | **58.899** | **45.408** |
| | 192 | 1.054 | 0.808 | **0.270** | **0.355** | **74.383** | **56.064** |
| | 336 | 1.523 | 0.948 | **0.304** | **0.374** | **80.039** | **60.549** |
| | 720 | 3.878 | 1.474 | **0.398** | **0.435** | **89.737** | **70.488** |
| Electricity | 96 | 0.336 | 0.416 | **0.324** | **0.407** | **3.571** | **2.163** |
| | 192 | 0.360 | 0.441 | **0.336** | **0.419** | **6.667** | **4.989** |
| | 336 | 0.356 | 0.439 | **0.347** | **0.429** | **2.528** | **2.278** |
| | 720 | 0.386 | 0.452 | **0.381** | **0.448** | **1.295** | **0.885** |
| Exchange | 96 | 1.029 | 0.809 | **0.512** | **0.569** | **50.243** | **29.666** |
| | 192 | 1.155 | 0.867 | **0.791** | **0.696** | **31.515** | **19.723** |
| | 336 | 1.589 | 1.011 | **1.040** | **0.763** | **34.550** | **24.530** |
| | 720 | 3.011 | 1.431 | **1.213** | **0.842** | **59.714** | **41.160** |
| Traffic | 96 | 0.744 | 0.420 | **0.723** | **0.408** | **2.823** | **2.857** |
| | 192 | 0.753 | 0.426 | **0.735** | **0.414** | **2.390** | **2.817** |
| | 336 | 0.876 | 0.495 | **0.811** | **0.462** | **7.420** | **6.667** |
| | 720 | 1.011 | 0.578 | **0.985** | **0.566** | **2.572** | **2.076** |
| Weather | 96 | 0.315 | 0.382 | **0.180** | **0.256** | **42.857** | **32.984** |
| | 192 | 0.428 | 0.449 | **0.253** | **0.331** | **40.888** | **26.281** |
| | 336 | 0.620 | 0.554 | **0.296** | **0.361** | **52.258** | **34.838** |
| | 720 | 0.975 | 0.722 | **0.392** | **0.426** | **59.795** | **40.997** |
| ILI | 24 | 5.349 | 1.582 | **5.313** | **1.559** | **0.673** | **1.454** |
| | 36 | **5.203** | **1.572** | 5.260 | 1.581 | -1.096 | -0.573 |
| | 48 | **5.286** | **1.594** | 5.415 | 1.623 | -2.440 | -1.819 |
| | 60 | 5.419 | 1.620 | **5.300** | **1.593** | **2.196** | **1.667** |

Table 7: Autoformer multivariate

| | | Autoformer | | TSAA | | | |
| | | MSE | MAE | MSE↓ | MAE↓ | MSE%↑ | MAE%↑ |
|---|---|---|---|---|---|---|---|
| ETTm2 | 96 | 0.231 | 0.310 | **0.211** | **0.293** | **8.658** | **5.484** |
| | 192 | 0.289 | 0.346 | **0.269** | **0.327** | **6.920** | **5.491** |
| | 336 | 0.341 | 0.375 | **0.322** | **0.359** | **5.572** | **4.267** |
| | 720 | 0.444 | 0.434 | **0.410** | **0.407** | **7.658** | **6.221** |
| Electricity | 96 | 0.200 | 0.316 | **0.188** | **0.302** | **6.000** | **4.430** |
| | 192 | **0.217** | **0.326** | 0.221 | 0.328 | -1.843 | -0.613 |
| | 336 | 0.258 | 0.356 | **0.252** | **0.352** | **2.326** | **1.124** |
| | 720 | 0.261 | 0.363 | **0.248** | **0.351** | **4.981** | **3.306** |
| Exchnage | 96 | 0.150 | 0.281 | **0.143** | **0.272** | **4.667** | **3.203** |
| | 192 | **0.318** | **0.409** | 0.344 | 0.425 | -8.176 | -3.912 |
| | 336 | **0.713** | **0.616** | 0.742 | 0.641 | -4.067 | -4.058 |
| | 720 | **1.246** | **0.872** | 1.997 | 1.170 | -60.273 | -34.174 |
| Traffic | 96 | 0.615 | 0.384 | **0.602** | **0.375** | **2.114** | **2.344** |
| | 192 | 0.670 | 0.421 | **0.663** | **0.416** | **1.045** | **1.188** |
| | 336 | 0.635 | 0.392 | **0.627** | **0.387** | **1.260** | **1.276** |
| | 720 | **0.658** | **0.402** | 0.662 | 0.405 | -0.608 | -0.746 |
| Weather | 96 | 0.259 | 0.332 | **0.216** | **0.292** | **16.602** | **12.048** |
| | 192 | 0.298 | 0.356 | **0.278** | **0.336** | **6.711** | **5.618** |
| | 336 | 0.357 | 0.394 | **0.341** | **0.381** | **4.482** | **3.299** |
| | 720 | 0.422 | 0.431 | **0.397** | **0.410** | **5.924** | **4.872** |
| ILI | 24 | **3.549** | **1.305** | 3.565 | **1.302** | -0.451 | **0.230** |
| | 36 | 2.834 | 1.094 | **2.754** | **1.068** | **2.823** | **2.377** |
| | 48 | 2.889 | 1.122 | **2.856** | **1.114** | **1.142** | **0.713** |
| | 60 | **2.818** | **1.118** | 2.826 | **1.118** | -0.284 | **0.000** |

Table 8: FEDformer-f multivariate

| | | FEDformer-f | | TSAA | | | |
| | | MSE | MAE | MSE↓ | MAE↓ | MSE%↑ | MAE%↑ |
|---|---|---|---|---|---|---|---|
| ETTm2 | 96 | 0.189 | 0.282 | **0.187** | **0.274** | **1.058** | **2.837** |
| | 192 | 0.258 | 0.326 | **0.255** | **0.314** | **1.163** | **3.681** |
| | 336 | 0.323 | 0.363 | **0.311** | **0.350** | **3.715** | **3.581** |
| | 720 | 0.425 | 0.421 | **0.406** | **0.403** | **4.471** | **4.276** |
| Electricity | 96 | **0.185** | **0.300** | 0.185 | 0.300 | 0.000 | 0.000 |
| | 192 | **0.201** | **0.316** | 0.201 | 0.316 | 0.000 | 0.000 |
| | 336 | **0.214** | **0.329** | 0.214 | 0.329 | 0.000 | 0.000 |
| | 720 | **0.246** | **0.353** | 0.246 | 0.353 | 0.000 | 0.000 |
| Exchange | 96 | **0.142** | **0.271** | 0.149 | 0.277 | -4.930 | -2.214 |
| | 192 | 0.278 | 0.383 | **0.273** | **0.380** | **1.799** | **0.783** |
| | 336 | **0.450** | **0.497** | 0.517 | 0.540 | -14.889 | -8.652 |
| | 720 | **1.181** | **0.841** | 2.440 | 1.343 | -106.605 | -59.691 |
| Traffic | 96 | **0.577** | **0.361** | 0.577 | 0.362 | **0.000** | -0.277 |
| | 192 | 0.610 | 0.379 | **0.601** | **0.371** | **1.475** | **2.111** |
| | 336 | 0.623 | 0.385 | **0.619** | **0.383** | **0.642** | **0.519** |
| | 720 | **0.632** | **0.388** | 0.632 | 0.388 | 0.000 | 0.000 |
| Weather | 96 | 0.236 | 0.316 | **0.207** | **0.285** | **12.288** | **9.810** |
| | 192 | 0.273 | 0.333 | **0.252** | **0.311** | **7.692** | **6.607** |
| | 336 | 0.332 | 0.371 | **0.313** | **0.355** | **5.723** | **4.313** |
| | 720 | 0.408 | 0.418 | **0.382** | **0.395** | **6.373** | **5.502** |
| ILI | 24 | 3.268 | 1.257 | **3.150** | **1.219** | **3.611** | **3.023** |
| | 36 | 2.648 | 1.068 | **2.578** | **1.049** | **2.644** | **1.779** |
| | 48 | 2.615 | 1.072 | **2.609** | **1.069** | **0.229** | **0.280** |
| | 60 | 2.866 | 1.158 | **2.805** | **1.140** | **2.128** | **1.554** |

Table 9: FEDformer-w multivariate

| | | FEDformer-w | | TSAA | | | |
| | | MSE | MAE | MSE↓ | MAE↓ | MSE%↑ | MAE%↑ |
|---|---|---|---|---|---|---|---|
| ETTm2 | 96 | 0.205 | 0.290 | **0.199** | **0.280** | **2.927** | **3.448** |
| | 192 | 0.270 | 0.329 | **0.255** | **0.314** | **5.556** | **4.559** |
| | 336 | 0.328 | 0.364 | **0.316** | **0.352** | **3.659** | **3.297** |
| | 720 | 0.433 | 0.425 | **0.410** | **0.404** | **5.312** | **4.941** |
| Electricity | 96 | 0.196 | 0.310 | **0.183** | **0.297** | **6.633** | **4.194** |
| | 192 | 0.199 | 0.310 | **0.195** | **0.309** | **2.010** | **0.323** |
| | 336 | 0.217 | 0.334 | **0.208** | **0.323** | **4.147** | **3.293** |
| | 720 | 0.248 | 0.357 | **0.238** | **0.348** | **4.032** | **2.521** |
| Exchange | 96 | 0.151 | 0.282 | **0.144** | **0.272** | **4.636** | **3.546** |
| | 192 | 0.284 | 0.391 | **0.270** | **0.378** | **4.930** | **3.325** |
| | 336 | **0.442** | **0.493** | 0.459 | 0.504 | -3.846 | -2.231 |
| | 720 | **1.227** | **0.868** | 1.952 | 1.167 | -59.087 | -34.447 |
| Traffic | 96 | 0.584 | 0.368 | **0.565** | **0.352** | **3.253** | **4.348** |
| | 192 | 0.596 | 0.375 | **0.571** | **0.351** | **4.195** | **6.400** |
| | 336 | 0.590 | 0.365 | **0.584** | **0.359** | **1.017** | **1.644** |
| | 720 | 0.613 | 0.375 | **0.607** | **0.368** | **0.979** | **1.867** |
| Weather | 96 | 0.269 | 0.347 | **0.213** | **0.291** | **20.818** | **16.138** |
| | 192 | 0.357 | 0.412 | **0.274** | **0.339** | **23.249** | **17.718** |
| | 336 | 0.422 | 0.456 | **0.335** | **0.379** | **20.616** | **16.886** |
| | 720 | 0.629 | **0.570** | **0.406** | **0.423** | **35.453** | **25.789** |
| ILI | 24 | **2.752** | 1.125 | 2.760 | **1.123** | -0.291 | **0.178** |
| | 36 | **2.318** | **0.980** | 2.362 | 0.984 | -1.898 | -0.408 |
| | 48 | 2.328 | 1.006 | **2.264** | **0.988** | **2.749** | **1.789** |
| | 60 | 2.574 | 1.081 | **2.520** | **1.062** | **2.098** | **1.758** |

Table 10: Informer univariate

| | | Informer | | TSAA | | | |
|---|---|---|---|---|---|---|---|
| | | MSE | MAE | MSE↓ | MAE↓ | MSE%↑ | MAE%↑ |
| ETTm2 | 96 | **0.085** | **0.225** | **0.085** | 0.226 | **0.000** | -0.444 |
| | 192 | 0.130 | 0.282 | **0.128** | **0.279** | **1.538** | **1.064** |
| | 336 | **0.161** | **0.314** | 0.164 | 0.317 | -1.863 | -0.955 |
| | 720 | 0.221 | 0.373 | **0.221** | **0.371** | **0.000** | **0.536** |
| Electricity | 96 | 0.261 | 0.367 | **0.245** | **0.356** | **6.130** | **2.997** |
| | 192 | **0.285** | **0.386** | 0.297 | 0.401 | -4.211 | -3.886 |
| | 336 | **0.324** | **0.417** | 0.433 | 0.485 | -33.642 | -16.307 |
| | 720 | **0.632** | **0.612** | 0.707 | 0.644 | -11.867 | -5.229 |
| Exchange | 96 | 0.490 | 0.554 | **0.149** | **0.307** | **69.592** | **44.585** |
| | 192 | 0.790 | 0.721 | **0.260** | **0.402** | **67.089** | **44.244** |
| | 336 | 2.146 | 1.223 | **0.744** | **0.671** | **65.331** | **45.135** |
| | 720 | 1.447 | 1.008 | **0.527** | **0.594** | **63.580** | **41.071** |
| Traffic | 96 | 0.262 | 0.348 | **0.254** | **0.342** | **3.053** | **1.724** |
| | 192 | 0.294 | 0.376 | **0.284** | **0.367** | **3.401** | **2.394** |
| | 336 | 0.308 | 0.390 | **0.300** | **0.384** | **2.597** | **1.538** |
| | 720 | 0.364 | 0.440 | **0.326** | **0.409** | **10.440** | **7.045** |
| Weather | 96 | 0.005 | 0.048 | **0.003** | **0.043** | **40.000** | **10.417** |
| | 192 | 0.004 | 0.051 | **0.003** | **0.038** | **25.000** | **25.490** |
| | 336 | **0.003** | **0.043** | 0.004 | 0.047 | -33.333 | -9.302 |
| | 720 | 0.004 | 0.049 | **0.003** | **0.043** | **25.000** | **12.245** |

Table 11: Autoformer univariate

| | | Autoformer | | TSAA | | | |
|---|---|---|---|---|---|---|---|
| | | MSE | MAE | MSE↓ | MAE↓ | MSE%↑ | MAE%↑ |
| ETTm2 | 96 | 0.123 | 0.270 | **0.106** | **0.250** | **13.821** | **7.407** |
| | 192 | 0.141 | 0.289 | **0.126** | **0.273** | **10.638** | **5.536** |
| | 336 | 0.170 | 0.319 | **0.139** | **0.290** | **18.235** | **9.091** |
| | 720 | 0.206 | 0.353 | **0.187** | **0.338** | **9.223** | **4.249** |
| Electricity | 96 | 0.454 | 0.508 | **0.388** | **0.456** | **14.537** | **10.236** |
| | 192 | 0.511 | 0.532 | **0.463** | **0.509** | **9.393** | **4.323** |
| | 336 | 0.739 | 0.651 | **0.493** | **0.517** | **33.288** | **20.584** |
| | 720 | 0.673 | 0.610 | **0.532** | **0.549** | **20.951** | **10.000** |
| Exchange | 96 | 0.149 | 0.308 | **0.146** | **0.295** | **2.013** | **4.221** |
| | 192 | **0.290** | **0.415** | 0.298 | 0.416 | -2.759 | -0.241 |
| | 336 | **0.708** | 0.662 | 0.712 | **0.643** | -0.565 | **2.870** |
| | 720 | **1.324** | **0.892** | 2.703 | 1.390 | -104.154 | -55.830 |
| Traffic | 96 | 0.266 | 0.372 | **0.248** | **0.351** | **6.767** | **5.645** |
| | 192 | 0.272 | 0.379 | **0.249** | **0.353** | **8.456** | **6.860** |
| | 336 | 0.261 | 0.374 | **0.239** | **0.347** | **8.429** | **7.219** |
| | 720 | 0.269 | 0.372 | **0.237** | **0.343** | **11.896** | **7.796** |
| Weather | 96 | 0.009 | 0.078 | **0.003** | **0.039** | **66.667** | **50.000** |
| | 192 | 0.009 | 0.068 | **0.003** | **0.040** | **66.667** | **41.176** |
| | 336 | 0.006 | 0.058 | **0.004** | **0.046** | **33.333** | **20.690** |
| | 720 | 0.007 | 0.063 | **0.004** | **0.046** | **42.857** | **26.984** |

Table 12: FEDformer-f univariate

| | | FEDformer-f | | TSAA | | | |
|---|---|---|---|---|---|---|---|
| | | MSE | MAE | MSE↓ | MAE↓ | MSE%↑ | MAE%↑ |
| ETTm2 | 96 | **0.068** | **0.198** | **0.068** | **0.198** | **0.000** | **0.000** |
| | 192 | **0.096** | **0.238** | **0.096** | **0.238** | **0.000** | **0.000** |
| | 336 | **0.138** | **0.286** | 0.140 | 0.290 | -1.449 | -1.399 |
| | 720 | **0.189** | **0.335** | 0.190 | 0.336 | -0.529 | -0.299 |
| Electricity | 96 | 0.244 | 0.364 | **0.244** | **0.356** | **0.000** | **2.198** |
| | 192 | **0.276** | 0.382 | 0.277 | **0.381** | -0.362 | **0.262** |
| | 336 | **0.347** | **0.432** | **0.347** | **0.432** | **0.000** | **0.000** |
| | 720 | **0.408** | **0.473** | **0.408** | **0.473** | **0.000** | **0.000** |
| Exchange | 96 | **0.133** | **0.284** | 0.145 | 0.293 | -9.023 | -3.169 |
| | 192 | **0.292** | **0.419** | 0.313 | 0.434 | -7.192 | -3.580 |
| | 336 | **0.477** | **0.532** | 0.575 | 0.595 | -20.545 | -11.842 |
| | 720 | **1.304** | **0.882** | 2.852 | 1.453 | -118.712 | -64.739 |
| Traffic | 96 | 0.210 | 0.318 | **0.191** | **0.292** | **9.048** | **8.176** |
| | 192 | 0.206 | 0.311 | **0.197** | **0.301** | **4.369** | **3.215** |
| | 336 | 0.217 | 0.322 | **0.204** | **0.309** | **5.991** | **4.037** |
| | 720 | 0.243 | 0.342 | **0.219** | **0.318** | **9.877** | **7.018** |
| Weather | 96 | 0.009 | 0.073 | **0.002** | **0.033** | **77.778** | **54.795** |
| | 192 | 0.007 | 0.067 | **0.002** | **0.034** | **71.429** | **49.254** |
| | 336 | 0.006 | 0.062 | **0.002** | **0.036** | **66.667** | **41.935** |
| | 720 | 0.006 | 0.060 | **0.002** | **0.039** | **66.667** | **35.000** |

Table 13: NBEATS-I univariate

| | | NBEATS-I | | TSAA | | | |
|---|---|---|---|---|---|---|---|
| | | MSE | MAE | MSE↓ | MAE↓ | MSE%↑ | MAE%↑ |
| ETTm2 | 96 | 0.080 | 0.213 | **0.075** | **0.202** | **6.250** | **5.164** |
| | 192 | 0.103 | 0.240 | **0.102** | **0.237** | **0.971** | **1.250** |
| | 336 | **0.162** | **0.312** | 0.171 | 0.321 | -5.556 | -2.885 |
| | 720 | **0.199** | **0.347** | 0.231 | 0.376 | -16.080 | -8.357 |
| Electricity | 96 | 0.326 | 0.402 | **0.270** | **0.360** | **17.178** | **10.448** |
| | 192 | 0.350 | 0.417 | **0.277** | **0.368** | **20.857** | **11.751** |
| | 336 | 0.393 | 0.440 | **0.310** | **0.394** | **21.120** | **10.455** |
| | 720 | 0.458 | 0.490 | **0.394** | **0.453** | **13.974** | **7.551** |
| Exchange | 96 | 0.210 | 0.344 | **0.093** | **0.238** | **55.714** | **30.814** |
| | 192 | 1.130 | 0.840 | **0.215** | **0.352** | **80.973** | **58.095** |
| | 336 | 1.587 | 1.047 | **0.532** | **0.572** | **66.478** | **45.368** |
| | 720 | 0.870 | 0.747 | **0.744** | **0.665** | **14.483** | **10.977** |
| Traffic | 96 | **0.181** | **0.268** | 0.183 | 0.270 | -1.105 | -0.746 |
| | 192 | 0.177 | 0.263 | **0.176** | **0.263** | **0.565** | **0.000** |
| | 336 | **0.180** | **0.271** | **0.180** | 0.270 | **0.000** | 0.369 |
| | 720 | 0.226 | 0.316 | **0.224** | **0.311** | **0.885** | **1.582** |
| Weather | 96 | 0.003 | 0.044 | **0.001** | **0.024** | **66.667** | **45.455** |
| | 192 | 0.004 | 0.046 | **0.001** | **0.027** | **75.000** | **41.304** |
| | 336 | 0.004 | 0.048 | **0.004** | **0.035** | **0.000** | **27.083** |
| | 720 | 0.004 | 0.049 | **0.002** | **0.034** | **50.000** | **30.612** |

Table 14: NBEATS-G univariate

| | | NBEATS-G | | TSAA | | | |
|---|---|---|---|---|---|---|---|
| | | MSE | MAE | MSE↓ | MAE↓ | MSE%↑ | MAE%↑ |
| ETTm2 | 96 | 0.080 | 0.210 | **0.071** | **0.192** | **11.250** | **8.571** |
| | 192 | 0.110 | 0.250 | **0.109** | **0.246** | **0.909** | **1.600** |
| | 336 | **0.172** | **0.320** | 0.176 | 0.323 | -2.326 | -0.938 |
| | 720 | **0.201** | **0.353** | 0.218 | 0.366 | -8.458 | -3.683 |
| Electricity | 96 | 0.324 | 0.397 | **0.263** | **0.354** | **18.827** | **10.831** |
| | 192 | 0.363 | 0.420 | **0.278** | **0.368** | **23.416** | **12.381** |
| | 336 | 0.392 | 0.443 | **0.350** | **0.422** | **10.714** | **4.740** |
| | 720 | 0.489 | 0.502 | **0.378** | **0.447** | **22.699** | **10.956** |
| Exchange | 96 | 0.223 | 0.351 | **0.093** | **0.236** | **58.296** | **32.764** |
| | 192 | 0.783 | 0.675 | **0.215** | **0.352** | **72.542** | **47.852** |
| | 336 | 2.622 | 1.266 | **1.167** | **0.861** | **55.492** | **31.991** |
| | 720 | 2.588 | 1.303 | **1.687** | **1.033** | **34.815** | **20.721** |
| Traffic | 96 | 0.159 | 0.240 | **0.158** | **0.239** | **0.629** | **0.417** |
| | 192 | 0.181 | 0.264 | **0.160** | **0.243** | **11.602** | **7.955** |
| | 336 | **0.155** | **0.239** | 0.156 | 0.244 | -0.645 | -2.092 |
| | 720 | 0.212 | 0.304 | **0.189** | **0.279** | **10.849** | **8.224** |
| Weather | 96 | 0.003 | 0.043 | **0.002** | **0.031** | **33.333** | **27.907** |
| | 192 | 0.004 | 0.047 | **0.002** | **0.028** | **50.000** | **40.426** |
| | 336 | 0.005 | 0.054 | **0.004** | **0.037** | **20.000** | **31.481** |
| | 720 | 0.004 | 0.048 | **0.002** | **0.036** | **50.000** | **25.000** |

## G.4 MAIN RESULTS WITH STANDARD DEVIATION

The following tables Tabs. 15, 16 augment the tables presented in Sec. 5 in the main text with standard deviation measures computed over three different seed numbers.

Table 15: Multivariate long-term time-series forecasting results with the standard deviation.

| | | Informer | | Autoformer | | FEDformer-w | | FEDformer-f | |
|---|---|---|---|---|---|---|---|---|---|
| | | MSE | MAE | MSE | MAE | MSE | MAE | MSE | MAE |
| ETTm2 | 96 | 0.545 ± 0.024 | 0.588 ± 0.014 | 0.231 ± 0.004 | 0.31 ± 0.0 | 0.205 ± 0.001 | 0.29 ± 0.0 | 0.189 ± 0.001 | 0.282 ± 0.001 |
| | 192 | 1.054 ± 0.044 | 0.808 ± 0.018 | 0.289 ± 0.013 | 0.346 ± 0.01 | 0.27 ± 0.0 | 0.329 ± 0.0 | 0.258 ± 0.002 | 0.326 ± 0.001 |
| | 336 | 1.523 ± 0.036 | 0.948 ± 0.014 | 0.341 ± 0.008 | 0.375 ± 0.004 | 0.328 ± 0.0 | 0.364 ± 0.0 | 0.323 ± 0.004 | 0.363 ± 0.004 |
| | 720 | 3.878 ± 0.072 | 1.474 ± 0.004 | 0.444 ± 0.005 | 0.434 ± 0.005 | 0.433 ± 0.005 | 0.425 ± 0.005 | 0.425 ± 0.008 | 0.421 ± 0.001 |
| Electricity | 96 | 0.336 ± 0.013 | 0.416 ± 0.007 | 0.2 ± 0.007 | 0.316 ± 0.007 | 0.196 ± 0.002 | 0.31 ± 0.002 | 0.185 ± 0.001 | 0.3 ± 0.001 |
| | 192 | 0.36 ± 0.015 | 0.441 ± 0.012 | 0.217 ± 0.0 | 0.326 ± 0.002 | 0.199 ± 0.001 | 0.31 ± 0.002 | 0.201 ± 0.007 | 0.316 ± 0.008 |
| | 336 | 0.356 ± 0.007 | 0.439 ± 0.006 | 0.258 ± 0.024 | 0.356 ± 0.013 | 0.217 ± 0.001 | 0.334 ± 0.001 | 0.214 ± 0.002 | 0.329 ± 0.002 |
| | 720 | 0.386 ± 0.012 | 0.452 ± 0.01 | 0.261 ± 0.002 | 0.363 ± 0.004 | 0.248 ± 0.004 | 0.357 ± 0.004 | 0.246 ± 0.002 | 0.353 ± 0.0 |
| Exchange | 96 | 1.029 ± 0.038 | 0.809 ± 0.02 | 0.15 ± 0.005 | 0.281 ± 0.005 | 0.151 ± 0.004 | 0.282 ± 0.004 | 0.142 ± 0.004 | 0.271 ± 0.004 |
| | 192 | 1.155 ± 0.038 | 0.867 ± 0.015 | 0.318 ± 0.012 | 0.409 ± 0.008 | 0.284 ± 0.003 | 0.391 ± 0.002 | 0.278 ± 0.003 | 0.383 ± 0.001 |
| | 336 | 1.589 ± 0.048 | 1.011 ± 0.011 | 0.713 ± 0.516 | 0.616 ± 0.243 | 0.442 ± 0.003 | 0.493 ± 0.001 | 0.45 ± 0.003 | 0.497 ± 0.001 |
| | 720 | 3.011 ± 0.302 | 1.431 ± 0.067 | 1.246 ± 0.007 | 0.872 ± 0.002 | 1.227 ± 0.024 | 0.868 ± 0.011 | 1.181 ± 0.018 | 0.841 ± 0.002 |
| Traffic | 96 | 0.744 ± 0.006 | 0.42 ± 0.006 | 0.615 ± 0.015 | 0.384 ± 0.009 | 0.584 ± 0.005 | 0.368 ± 0.005 | 0.577 ± 0.001 | 0.361 ± 0.002 |
| | 192 | 0.753 ± 0.01 | 0.426 ± 0.011 | 0.67 ± 0.068 | 0.421 ± 0.045 | 0.594 ± 0.007 | 0.372 ± 0.007 | 0.61 ± 0.003 | 0.379 ± 0.004 |
| | 336 | 0.876 ± 0.024 | 0.495 ± 0.013 | 0.635 ± 0.027 | 0.392 ± 0.014 | 0.59 ± 0.001 | 0.365 ± 0.002 | 0.623 ± 0.005 | 0.385 ± 0.004 |
| | 720 | 1.011 ± 0.032 | 0.578 ± 0.018 | 0.658 ± 0.011 | 0.402 ± 0.01 | 0.613 ± 0.006 | 0.375 ± 0.003 | 0.632 ± 0.006 | 0.388 ± 0.007 |
| Weather | 96 | 0.315 ± 0.004 | 0.382 ± 0.004 | 0.259 ± 0.014 | 0.332 ± 0.011 | 0.269 ± 0.011 | 0.347 ± 0.008 | 0.236 ± 0.024 | 0.316 ± 0.028 |
| | 192 | 0.428 ± 0.005 | 0.449 ± 0.012 | 0.298 ± 0.012 | 0.356 ± 0.012 | 0.357 ± 0.001 | 0.412 ± 0.002 | 0.273 ± 0.013 | 0.333 ± 0.012 |
| | 336 | 0.62 ± 0.027 | 0.554 ± 0.009 | 0.357 ± 0.01 | 0.394 ± 0.011 | 0.422 ± 0.008 | 0.456 ± 0.006 | 0.332 ± 0.016 | 0.371 ± 0.016 |
| | 720 | 0.975 ± 0.035 | 0.722 ± 0.009 | 0.422 ± 0.011 | 0.431 ± 0.01 | 0.629 ± 0.008 | 0.57 ± 0.004 | 0.408 ± 0.009 | 0.418 ± 0.008 |
| ILI | 24 | 5.349 ± 0.229 | 1.582 ± 0.052 | 3.549 ± 0.33 | 1.305 ± 0.068 | 2.752 ± 0.023 | 1.125 ± 0.005 | 3.268 ± 0.04 | 1.257 ± 0.014 |
| | 36 | 5.203 ± 0.129 | 1.572 ± 0.029 | 2.834 ± 0.166 | 1.094 ± 0.032 | 2.318 ± 0.017 | 0.98 ± 0.006 | 2.648 ± 0.034 | 1.068 ± 0.009 |
| | 48 | 5.286 ± 0.049 | 1.594 ± 0.018 | 2.889 ± 0.178 | 1.122 ± 0.033 | 2.328 ± 0.043 | 1.006 ± 0.013 | 2.615 ± 0.019 | 1.072 ± 0.004 |
| | 60 | 5.419 ± 0.103 | 1.62 ± 0.019 | 2.818 ± 0.157 | 1.118 ± 0.039 | 2.574 ± 0.052 | 1.081 ± 0.016 | 2.866 ± 0.031 | 1.158 ± 0.008 |

| | | TSAA | |
|---|---|---|---|
| | | MSE | MAE |
| ETTm2 | 96 | 0.187 ± 0.000 | 0.274 ± 0.000 |
| | 192 | 0.255 ± 0.001 | 0.314 ± 0.001 |
| | 336 | 0.304 ± 0.009 | 0.350 ± 0.002 |
| | 720 | 0.398 ± 0.007 | 0.403 ± 0.002 |
| Electricity | 96 | 0.183 ± 0.001 | 0.297 ± 0.001 |
| | 192 | 0.195 ± 0.001 | 0.309 ± 0.002 |
| | 336 | 0.208 ± 0.002 | 0.323 ± 0.002 |
| | 720 | 0.238 ± 0.003 | 0.348 ± 0.003 |
| Exchange | 96 | 0.143 ± 0.012 | 0.272 ± 0.012 |
| | 192 | 0.27 ± 0.002 | 0.378 ± 0.002 |
| | 336 | 0.459 ± 0.007 | 0.504 ± 0.007 |
| | 720 | 1.213 ± 0.056 | 0.842 ± 0.008 |
| Traffic | 96 | 0.565 ± 0.005 | 0.352 ± 0.004 |
| | 192 | 0.572 ± 0.002 | 0.351 ± 0.001 |
| | 336 | 0.584 ± 0.004 | 0.359 ± 0.005 |
| | 720 | 0.607 ± 0.002 | 0.368 ± 0.002 |
| Weather | 96 | 0.18 ± 0.024 | 0.256 ± 0.024 |
| | 192 | 0.252 ± 0.001 | 0.311 ± 0.002 |
| | 336 | 0.296 ± 0.001 | 0.355 ± 0.005 |
| | 720 | 0.382 ± 0.006 | 0.395 ± 0.007 |
| ILI | 24 | 2.76 ± 0.063 | 1.123 ± 0.016 |
| | 36 | 2.362 ± 0.024 | 0.984 ± 0.008 |
| | 48 | 2.264 ± 0.074 | 0.988 ± 0.012 |
| | 60 | 2.52 ± 0.064 | 1.062 ± 0.022 |

Table 16: Univariate long-term time-series forecasting results with the standard deviation.

| | | Informer | | Autoformer | | FEDformer-f | | N-BEATS-I | |
|---|---|---|---|---|---|---|---|---|---|
| | | MSE | MAE | MSE | MAE | MSE | MAE | MSE | MAE |
| ETTm2 | 96 | $0.085 \pm 0.004$ | $0.225 \pm 0.006$ | $0.123 \pm 0.017$ | $0.27 \pm 0.018$ | $0.068 \pm 0.001$ | $0.198 \pm 0.002$ | $0.08 \pm 0.003$ | $0.213 \pm 0.006$ |
| | 192 | $0.13 \pm 0.007$ | $0.282 \pm 0.008$ | $0.141 \pm 0.01$ | $0.289 \pm 0.01$ | $0.096 \pm 0.001$ | $0.238 \pm 0.001$ | $0.103 \pm 0.004$ | $0.24 \pm 0.006$ |
| | 336 | $0.161 \pm 0.008$ | $0.314 \pm 0.006$ | $0.17 \pm 0.046$ | $0.319 \pm 0.042$ | $0.138 \pm 0.013$ | $0.286 \pm 0.014$ | $0.162 \pm 0.009$ | $0.312 \pm 0.009$ |
| | 720 | $0.221 \pm 0.006$ | $0.373 \pm 0.007$ | $0.206 \pm 0.02$ | $0.353 \pm 0.017$ | $0.189 \pm 0.002$ | $0.335 \pm 0.002$ | $0.199 \pm 0.007$ | $0.347 \pm 0.007$ |
| Electricity | 96 | $0.261 \pm 0.005$ | $0.367 \pm 0.002$ | $0.454 \pm 0.014$ | $0.508 \pm 0.014$ | $0.244 \pm 0.001$ | $0.364 \pm 0.002$ | $0.326 \pm 0.006$ | $0.402 \pm 0.004$ |
| | 192 | $0.285 \pm 0.006$ | $0.386 \pm 0.003$ | $0.511 \pm 0.05$ | $0.532 \pm 0.027$ | $0.276 \pm 0.004$ | $0.382 \pm 0.004$ | $0.35 \pm 0.008$ | $0.417 \pm 0.005$ |
| | 336 | $0.324 \pm 0.004$ | $0.417 \pm 0.004$ | $0.739 \pm 0.086$ | $0.651 \pm 0.042$ | $0.347 \pm 0.007$ | $0.432 \pm 0.006$ | $0.393 \pm 0.008$ | $0.44 \pm 0.003$ |
| | 720 | $0.632 \pm 0.049$ | $0.612 \pm 0.028$ | $0.673 \pm 0.082$ | $0.61 \pm 0.037$ | $0.408 \pm 0.025$ | $0.473 \pm 0.015$ | $0.458 \pm 0.008$ | $0.49 \pm 0.002$ |
| Exchange | 96 | $0.49 \pm 0.065$ | $0.554 \pm 0.034$ | $0.149 \pm 0.004$ | $0.308 \pm 0.006$ | $0.133 \pm 0.015$ | $0.284 \pm 0.018$ | $0.21 \pm 0.047$ | $0.344 \pm 0.036$ |
| | 192 | $0.79 \pm 0.039$ | $0.721 \pm 0.016$ | $0.29 \pm 0.005$ | $0.415 \pm 0.004$ | $0.292 \pm 0.002$ | $0.419 \pm 0.003$ | $1.13 \pm 0.392$ | $0.84 \pm 0.153$ |
| | 336 | $2.146 \pm 0.25$ | $1.223 \pm 0.084$ | $0.708 \pm 0.108$ | $0.662 \pm 0.053$ | $0.477 \pm 0.002$ | $0.532 \pm 0.002$ | $1.587 \pm 0.219$ | $1.047 \pm 0.077$ |
| | 720 | $1.447 \pm 0.105$ | $1.008 \pm 0.038$ | $1.324 \pm 0.005$ | $0.892 \pm 0.002$ | $1.304 \pm 0.003$ | $0.882 \pm 0.0$ | $0.87 \pm 0.088$ | $0.747 \pm 0.015$ |
| Traffic | 96 | $0.262 \pm 0.006$ | $0.348 \pm 0.006$ | $0.266 \pm 0.005$ | $0.372 \pm 0.01$ | $0.21 \pm 0.006$ | $0.318 \pm 0.009$ | $0.181 \pm 0.004$ | $0.268 \pm 0.005$ |
| | 192 | $0.294 \pm 0.004$ | $0.376 \pm 0.006$ | $0.272 \pm 0.014$ | $0.379 \pm 0.01$ | $0.206 \pm 0.01$ | $0.311 \pm 0.006$ | $0.177 \pm 0.001$ | $0.263 \pm 0.001$ |
| | 336 | $0.308 \pm 0.007$ | $0.39 \pm 0.002$ | $0.261 \pm 0.016$ | $0.374 \pm 0.016$ | $0.217 \pm 0.005$ | $0.322 \pm 0.0$ | $0.18 \pm 0.005$ | $0.271 \pm 0.006$ |
| | 720 | $0.364 \pm 0.018$ | $0.44 \pm 0.017$ | $0.269 \pm 0.012$ | $0.372 \pm 0.005$ | $0.243 \pm 0.021$ | $0.342 \pm 0.021$ | $0.226 \pm 0.003$ | $0.316 \pm 0.004$ |
| Weather | 96 | $0.005 \pm 0.001$ | $0.048 \pm 0.005$ | $0.009 \pm 0.002$ | $0.078 \pm 0.009$ | $0.009 \pm 0.004$ | $0.073 \pm 0.014$ | $0.003 \pm 0.001$ | $0.044 \pm 0.006$ |
| | 192 | $0.004 \pm 0.0$ | $0.051 \pm 0.001$ | $0.009 \pm 0.002$ | $0.068 \pm 0.001$ | $0.007 \pm 0.002$ | $0.067 \pm 0.008$ | $0.004 \pm 0.0$ | $0.046 \pm 0.003$ |
| | 336 | $0.003 \pm 0.001$ | $0.043 \pm 0.004$ | $0.006 \pm 0.001$ | $0.058 \pm 0.006$ | $0.006 \pm 0.001$ | $0.062 \pm 0.007$ | $0.004 \pm 0.001$ | $0.048 \pm 0.004$ |
| | 720 | $0.004 \pm 0.002$ | $0.049 \pm 0.007$ | $0.007 \pm 0.001$ | $0.063 \pm 0.004$ | $0.006 \pm 0.001$ | $0.06 \pm 0.007$ | $0.004 \pm 0.0$ | $0.049 \pm 0.003$ |

| | | N-BEATS-G | | TSAA | |
|---|---|---|---|---|---|
| | | MSE | MAE | MSE | MAE |
| ETTm2 | 96 | $0.08 \pm 0.005$ | $0.21 \pm 0.007$ | $0.068 \pm 0.001$ | $0.192 \pm 0.002$ |
| | 192 | $0.11 \pm 0.004$ | $0.25 \pm 0.005$ | $0.096 \pm 0.001$ | $0.237 \pm 0.004$ |
| | 336 | $0.172 \pm 0.007$ | $0.32 \pm 0.007$ | $0.139 \pm 0.005$ | $0.29 \pm 0.005$ |
| | 720 | $0.201 \pm 0.008$ | $0.353 \pm 0.008$ | $0.187 \pm 0.008$ | $0.336 \pm 0.001$ |
| Electricity | 96 | $0.324 \pm 0.005$ | $0.397 \pm 0.002$ | $0.244 \pm 0.006$ | $0.354 \pm 0.012$ |
| | 192 | $0.363 \pm 0.005$ | $0.42 \pm 0.003$ | $0.277 \pm 0.003$ | $0.368 \pm 0.003$ |
| | 336 | $0.392 \pm 0.002$ | $0.443 \pm 0.006$ | $0.31 \pm 0.006$ | $0.394 \pm 0.005$ |
| | 720 | $0.489 \pm 0.013$ | $0.502 \pm 0.005$ | $0.378 \pm 0.026$ | $0.447 \pm 0.012$ |
| Exchange | 96 | $0.223 \pm 0.046$ | $0.351 \pm 0.045$ | $0.093 \pm 0.008$ | $0.236 \pm 0.007$ |
| | 192 | $0.783 \pm 0.203$ | $0.675 \pm 0.085$ | $0.215 \pm 0.035$ | $0.352 \pm 0.017$ |
| | 336 | $2.622 \pm 1.07$ | $1.266 \pm 0.23$ | $0.532 \pm 0.045$ | $0.572 \pm 0.006$ |
| | 720 | $2.588 \pm 0.11$ | $1.303 \pm 0.019$ | $0.527 \pm 0.047$ | $0.594 \pm 0.019$ |
| Traffic | 96 | $0.159 \pm 0.001$ | $0.24 \pm 0.002$ | $0.158 \pm 0.001$ | $0.239 \pm 0.001$ |
| | 192 | $0.181 \pm 0.005$ | $0.264 \pm 0.001$ | $0.16 \pm 0.0$ | $0.243 \pm 0.001$ |
| | 336 | $0.155 \pm 0.001$ | $0.239 \pm 0.001$ | $0.156 \pm 0.004$ | $0.244 \pm 0.006$ |
| | 720 | $0.212 \pm 0.003$ | $0.304 \pm 0.003$ | $0.189 \pm 0.002$ | $0.279 \pm 0.003$ |
| Weather | 96 | $0.003 \pm 0.0$ | $0.043 \pm 0.002$ | $0.001 \pm 0.0$ | $0.024 \pm 0.0$ |
| | 192 | $0.004 \pm 0.001$ | $0.047 \pm 0.004$ | $0.001 \pm 0.0$ | $0.027 \pm 0.001$ |
| | 336 | $0.005 \pm 0.001$ | $0.054 \pm 0.004$ | $0.002 \pm 0.0$ | $0.035 \pm 0.004$ |
| | 720 | $0.004 \pm 0.0$ | $0.048 \pm 0.002$ | $0.002 \pm 0.0$ | $0.034 \pm 0.002$ |

