# OpenReview forum: "Time-Series AutoAugment: Data Augmentation Policy Search for Long-Term Forecasting"
_ICLR.cc/2024/Conference — Submitted to ICLR 2024_

### Official Review · Reviewer_dM8C · 2023-10-30

**Soundness:** 2 fair
**Presentation:** 3 good
**Contribution:** 2 fair
**Rating:** 5
**Confidence:** 4

**Summary:**

I have already reviewed this paper for another conference and I could not find substantial changes in the paper and/or presented method. Hence, I will repeat my summary and some of my main points of the other review. I would kindly ask the authors to point out if I missed substantial updates of the paper.

This paper introduces a data augmentation method for time series forecasting (specifically long-term forecasting). The proposed method starts from a partially trained base model and then searches over a given augmentation space for a good augmentation policy with Bayesian optimization. The method alternates between policy selection (using the Tree-Structured Parzen Estimator) and fine tuning the models with Asynchronous Successive Halving to reduce the required computation. The authors evaluate the method on multivariate and univariate forecasting tasks using several transformer variants and N-BEATS and find that the proposed method reduces the MSE/MAE over these base models (and reduces the error further than random augmentation). In addition, the authors present which policies have been selected by their method and perform an analysis on the impact of the selected hyperparameters of their model.

**Strengths:**

### Originality

The authors develop an automated method for data augmentation policy selection for time series forecasting. While the individual components (augmentations, the Bayesian Optimization procedure) are not novel, the combination and application to time series forecasting has not been proposed before (to my knowledge). The application of Bayesian Optimization to select policies has been proposed before (Zhang, Cui, Yang 2019, https://arxiv.org/abs/1905.02610).

### Quality

The paper analyses several baselines and datasets in the univariate and multivariate setting. The authors also provide analysis on parameter choices and the selected policies for their method, which is interesting. I think the evaluation has some major weaknesses, which I will discuss in the Weaknesses section.

### Significance

The proposed method is an interesting finding for time series forecasting, but the significance for adjacent fields is limited because the authors do not develop any novel technique but rather adapt existing techniques for time series forecasting.

**Weaknesses:**

One weakness of the paper is the choice of baselines. One interesting observation is that the largest error reductions are observed for the Exchange and Weather datasets. Given that Exchange is a dataset from the finance domain it likely shows a random-walk like behaviour. Weather has 10 minute recordings of meteorological measurements, so I assume that this data likely has a large random component because the forecasted horizon is probably too small to capture seasonality over the year (5 days for the 720 time step horizon (720/(24*6))=5). This is in contrast to the small improvements observed for the other datasets that have a strong seasonality component (like Electricity, Traffic, and ILI). Thus, I’m wondering whether the main impact of the data augmentation mostly improves the forecasts for random-walk data and how this compares to baselines that are more suitable for this kind of data. I would kindly ask to the authors to include baselines in their evaluation that are suited for random-walk data. Two suggestions are the naive method proposed in Bergmeir et al., KDD 2022 (https://link.springer.com/article/10.1007/s10618-022-00894-5) and exponential smoothing. For completeness, it would be interesting for the reader to understand what the gain is of using complex transformer models plus sophisticated data augmentation relative to rather simple baselines for seasonal data. Comparing to DHR-ARIMA (also mentioned in that paper) for seasonal methods would be interesting (or D-Linear, which the authors cite in their introduction, seasonal naive for a baseline for seasonal time series).

Given that some of the improvements are small, I would suggest a different way of presenting the data that includes a significant test and takes the variance of re-running the models with different seeds into account. I suggest to use critical diagrams for that (Demšar, JMLR 2006, https://jmlr.org/papers/v7/demsar06a.html). This would make it easier for the reader to judge the significance of the error reduction that is achieved here.

**Questions:**

These are my main questions as explained in the Weaknesses section:

How do simple baselines suitable for random walks (naive method and exponential smoothing) compare in this evaluation?

What is the significance of the improvements of TSAA when presenting the results with a critical diagram that includes a statistical test (Demšar, JMLR 2006, https://jmlr.org/papers/v7/demsar06a.html)?

I would consider increasing my score if these questions are addressed.

---

### Official Review · Reviewer_dahn · 2023-11-01

**Soundness:** 3 good
**Presentation:** 3 good
**Contribution:** 2 fair
**Rating:** 5
**Confidence:** 4

**Summary:**

This paper presents an automatic time series augmentation (TSAA) search approach formulated as a bilevel optimization problem. TSAA uses Tree-structured Parzen Estimator with Expected Improvement as a surrogate model to search for promising augmentation strategies. To avoid training from scratch each time given an augmentation policy, TSAA finetunes from a set of shared weights which is got from training without augmentation. Further, TSAA uses Asynchronous Successive Halving to discard unpromising runs to improve the optimization efficiency.

**Strengths:**

1. Data augmentation is much less explored for time series problems than other data modalities. This work presents a meaningful way to automatically search for useful augmentation strategies on different datasets.
2. The writing is clear and easy to follow with good visualizations.

**Weaknesses:**

1. From Table 6-14, it can be observed that under a lot of settings, TSAA even makes the performance much worse than the baseline, especially for some specific datasets, e.g. Exchange. I did not find the interpretation of such results, which makes it a bit hard for the community to understand when TSAA could fail.
2. TSAA is mainly evaluated on Transformer-based models. As a data augmentation approach, we would expect it to be agnostic to the model architecture. The effectiveness of TSAA on more recent MLP-based architecture [1] or convolution-based architecture [2] is not shown.
3. Overall, TSAA seems to be an application of multiple well-established approaches (e.g., TPE estimator and ASHA) in hyperparameter optimization and time series augmentation. While these approaches were not adapted to time-series modality before, I did not find enough takeaway insights from applying these approaches to time-series data.

[1] Are Transformers Effective for Time Series Forecasting? AAAI 2023
[2] TimesNet: Temporal 2D-Variation Modeling for General Time Series Analysis. ICLR 2023

**Questions:**

1. In Figure 4, Flip and Seasonality Downscale is used for NBEATS-G. Is there some way to understand why prediction + TSAA fixes the underestimated prediction in this case?
2. It is claimed that Trend Downscale is frequently used for multiple datasets, which is probably because deep models tend to overestimate. Based on such a claim, I would expect to see some examples in Figure 4.
3. What is the compute time to apply TSAA compared with vanilla training? It seems that the first half of training time remains the same (since $\beta=0.5$), but the second half would involve an expensive search process and significantly increase the compute time of training.
4. How do interpret the performance degradation on a specific dataset, e.g. Exchange?

---

### Official Review · Reviewer_E2oo · 2023-11-01

**Soundness:** 3 good
**Presentation:** 4 excellent
**Contribution:** 3 good
**Rating:** 5
**Confidence:** 4

**Summary:**

This paper propose a method called Time-Series AutoAugment (TSAA) that efficiently searches for an optimal data augmentation policy. The method involves a two-step process: a partial train of the non-augmented model followed by an iterative split process.The iterative process alternates between finding a good augmentation policy using Bayesian optimization and fine-tuning the model while pruning poor runs. The authors evaluate TSAA on challenging univariate and multivariate forecasting benchmark problems and show that it outperforms several strong baselines in most cases.
The main contributions of the work are three-fold:
1.Development of a novel automatic data augmentation approach for long-term time-series forecasting tasks
2.Analysis of the optimal policies found by the approach
3.Extensive evaluation on TSAA and benchmark problems.

**Strengths:**

1.Originality:While data augmentation techniques are commonly used in vision tasks, their application to time-series forecasting is relatively new. TSAA proposes a novel automatic DA method for TS data using a set of time-series transformations that manipulate certain features of the data while leaving others unchanged.This approach fills a gap in the field of time-series forecasting.
2.Quality:The paper demonstrates the effectiveness of TSAA through extensive experiments on multiple datasets and models. The authors compare TSAA with several baseline models and other automatic augmentation methods, providing a comprehensive evaluation of its performance. The experiments are conducted with multiple seed numbers on the same system and the results are averaged, ensuring robustness and reliability.
3.Clarity:The paper is well-written and organized, making it easy to follow the proposed approach and experimental results. The authors provide detailed explanations of the time-series transformations used in TSAA and the overview of TPE and ASHA ensuring a clear understanding of the augmentation process. The link between the task of automatic augmentation and a bi-level optimization makes the two-step process of TSAA better understood.
4.Significance:The paper addresses the challenge of data augmentation in time-series forecasting.By proposing TSAA, the authors offer a practical solution to improve the performance of time-series forecasting models. The extensive experiments demonstrate the effectiveness of TSAA in improving forecasting accuracy across multiple datasets and models. The findings of this research have the potential to benefit researchers and practitioners working on TSF tasks.

**Weaknesses:**

1.The paper does not extensively discuss the limitations of TSAA or potential challenges in its application. It would be beneficial to address any constraints or assumptions made in the proposed approach, as well as potential scenarios where TSAA may not be as effective.
2.The paper does not further explain the relationship of the dataset to its selected transformations.For instance,in the univariate time-series forecasting results,why Exchange need more Upscale but ECL need more Mixup?

**Questions:**

1.In the 5.2 section, we can see TSAA yeild stronger performance in the long-horizon settings.Can you give a more clear analysis and explanation for this experimental phenomenon?
2.In Table.1 and Table.2, some datasets can be highly enhance such as Weather,but some datasets yield very little gain such as ETTm2.How does this difference come about and is it due to the dataset? Or is TSAA a weekness?

**Details Of Ethics Concerns:**

Not applicable.

---

### Official Review · Reviewer_9xnW · 2023-11-04

**Soundness:** 2 fair
**Presentation:** 2 fair
**Contribution:** 1 poor
**Rating:** 3
**Confidence:** 5

**Summary:**

This paper studies an automatic data augmentation approach for time series forecasting. The main idea is to use bi-level optimization to learn weights as an augmentation policy from a designed dictionary of time-series transformations.

**Strengths:**

S1. The paper covers both univariate and multivariate time series forecasting tasks.

S2. Table 3 explores various time series transformations.

**Weaknesses:**

W1. The novelty of the proposed auto-augmented framework is not enough as it has been studied in image domains (Bayesian optimization [1] or bilevel/meta-learning methods).
>-  [1] 2019 Learning Optimal Data Augmentation Policies via Bayesian Optimization for Image Classification Tasks

W2. The paper's investigation of time series data augmentation is limited, as it does not fully explore existing learnable augmentation strategies, such as generative augmentation [1,2], adaptive weighting schemes [3], and gating-based selection [4]. These approaches are not discussed or compared in the paper.
>- [1] Two Birds with One Stone: Series Saliency for Accurate and Interpretable Multivariate Time Series Forecasting
>-  [2]  TSA-GAN: A Robust Generative Adversarial Networks for Time Series Augmentation
>-  [3] Adaptive Weighting Scheme for Automatic Time-Series Data Augmentation
>-  [4] 2022 Dynamic Data Augmentation with Gating Networks for Time Series Recognition

W3. I believe that data augmentation should be a general technique applicable to various domains/datasets/tasks, and thus, the paper's claims that less attention has been given to automatic augmentation of time-series problems, such as long-term forecasting, may not be entirely convincing. To strengthen the paper's argument, it would be beneficial to evaluate the proposed method on a wider range of time series tasks. Additionally, it is unclear how the optimal augmentation policy for forecasting differs from that for classification, and further clarification on this point would be helpful.

W4. Table 1 and Table 2 are misleading. It is unclear what base networks are trained with the auto-augment method (the col of TSAA). Moreover, it would be better to compare existing time series auto-augmentation methods (on various base ts models) in the experiment section.

W5. Table 5 in the Appendix compares only Fast AA and RandAugment, and I noticed that Fast AA performs significantly worse than the baseline on ETTm2, Traffic, and ILI. Could you provide some insights into the possible reasons for this difference?

W6. Unlike image transformations, selecting appropriate hyper-parameter ranges for time series transformations can be challenging. As demonstrated in Figure 5, many of the augmented samples are significantly different from the original time series. This raises the question of how to construct a dictionary of time-series transformations that is suitable for various tasks. Could you elaborate on this issue and provide more insights on how to design such a dictionary?

W7. Given that the data will be transformed and passed through the network multiple times (equivalent to the number of transformations), is there a significant increase in computational cost associated with this approach?

**Questions:**

Please see my questions in Weaknesses.

---

### Meta-Review · Area_Chair_uw2A · 2023-12-05

**Metareview:**

Automated data augmentation has been studied mostly for classification tasks in recent years. This paper studies it in the context of time-series forecasting which is relatively less studied and hence an important research topic. Nevertheless, the proposed framework is limited in novelty as it is mostly based on multiple existing techniques. On the experimental design, the choice of both models and datasets should be broadened to demonstrate the generality of the proposed method. Moreover, for some datasets, the performance gain is very small. The significance of the proposed method needs to be demonstrated more convincingly. In summary, while this work studies an important problem, its current form is below the acceptance standard of ICLR.

**Justification For Why Not Higher Score:**

It is well below the acceptance standard.

**Justification For Why Not Lower Score:**

N/A

---

### Decision · Program_Chairs · 2024-01-16

Reject